# Mechanistically driven transnidal hemodynamic manipulations enhance simulated endovascular transvenous treatments for brain AVMs

Tarik F. Massoud [1,2] ✉, Bryce C. Vu [3], Kellen Vo Vu [4], Jeremy J. Heit [1,5] & Siddhant Suri Dhawan [6]

## Abstract

**Background** Treatment strategies for plexiform brain arteriovenous malformations (bAVMs) have evolved to include minimally invasive transvenous embolization. Since its conceptualization as "transvenous retrograde nidus sclerotherapy under controlled hypotension" (TRENSH), this approach has shown curative potential for highly selected small bAVMs when using adhesive embolic agents. Further innovation is required to extend safety and efficacy to larger, more complex bAVMs. We conceived and evaluated a set of theoretical hemodynamic manipulations within the venous outflow of bAVMs designed to augment nidus retropermeation during simulated TRENSH.

**Methods** We used two complementary experimental platforms. First, we developed a computational bAVM model to simulate hemodynamic variations during TRENSH, including: (1) degrees of controlled arterial hypotension; (2) effects of temporary balloon occlusion of arterial feeders; (3) differing draining vein (DV) retrograde injection pressures; (4) use of alternative DVs for retroinjection; (5) elevation of central venous pressure (CVP) during injection; and (6) cardiac-cycle–synchronized diastolic retroinjection. Second, we used a carotid–jugular fistula rete mirabile AVM model in six pigs to evaluate combinations of induced hypotension and venous hypertension, simulating raised CVP to enhance retropermeation during TRENSH-like maneuvers.

**Results** Here we show that CVP elevation, retrograde injection through dominant DVs, maximally safe transvenous injection pressures, and a distinct strategy of synchronized diastolic-phase DV retroinjection each increase nidus retropermeation in experimental simulations.

**Conclusions** These theoretical TRENSH-derived venous manipulation strategies may offer adjunctive benefit for future transvenous treatment of large bAVMs. The findings provide a conceptual foundation for further validation studies to determine their translational feasibility and potential incorporation into advanced clinical TRENSH paradigms.

## Plain language summary

Brain arteriovenous malformations are abnormal tangles of blood vessels that can be difficult and risky to treat, especially when they are large or have complex structures. A current treatment approach works by injecting material backward into the veins that drain these tangles, but it is not yet effective for many patients. In this study, we explored alternative ways to improve this treatment by adjusting blood flow conditions during the procedure. We used computer models and an animal model to test how changes in blood pressure and timing of injections might help treatment spread more fully through the abnormal vessels. These early results suggest different strategies could eventually make this minimally invasive treatment safer and effective for more people.

Plexiform bAVMs are potentially life-threatening congenital cerebrovascular anomalies consisting of fistulous connections between dysplastic arterial feeders (AFs) and draining veins (DVs) with an intervening central fast-flowing nidus composed of tangled, fragile, low-resistance microvessels[1–4]. Unruptured bAVMs have a 1.3–3% annual rate of hemorrhage[3], and are the commonest cause of non-traumatic intracranial

[1]Division of Neuroimaging and Neurointervention, Department of Radiology, Stanford University School of Medicine, Stanford, CA, USA. [2]Division of Interventional Neuroradiology, Department of Radiological Sciences, David Geffen School of Medicine at UCLA, Los Angeles, CA, USA. [3]Arizona State University, Tempe, AZ, USA. [4]MD Program, Weill Cornell Medicine, New York, NY, USA. [5]Department of Neurosurgery, Stanford University School of Medicine, Stanford, CA, USA. [6]Department of Bioengineering, Stanford University Schools of Engineering and Medicine, Stanford, CA, USA. ✉e-mail: tmassoud@stanford.edu

hemorrhage in the 15 to 45-year age group[1]. Owing to a serious lifelong hemorrhagic risk and related morbidity and mortality, definitive treatment to obliterate the entire nidus of a bAVM is the usual goal[5,6]. However, complex and variable angioarchitectures, strong arteriovenous shunting of blood, and frequent vicinity to eloquent brain parenchyma, make bAVMs among the most challenging vascular pathologies to treat and achieve cure[2]. Current multimodality treatment options include microsurgical resection, minimally invasive endovascular embolization, and radiosurgery, used alone or in combination[2,4]. The knowledge gained from trials such as TOBAS (Treatment of Brain Arteriovenous Malformations Study) will provide a better understanding of safety and efficacy for different bAVM treatments[7]. Endovascular therapy, in particular, is constantly evolving and now includes not only transarterial[8], but also transvenous or combined treatment approaches[4].

Transvenous bAVM embolization is a relatively new technique[2,9–13], first conceptualized by Massoud and Hademenos in 1999 as "transvenous retrograde nidus sclerotherapy under controlled hypotension" (TRENSH)[14]. Although the prior decade to that had seen significant success with transvenous treatments of dural arteriovenous fistulas[15], those approaches were not extrapolated to bAVMs. The underlying rationale for TRENSH dictates that temporary controlled systemic hypotension, with or without the aid of temporary balloon occlusion (TBO) of major bAVM AFs, could yield sufficient intranidal flow stasis to allow the possibility of transvenous retrograde nidus injection with a sclerosant (a "chemical" or "biological/molecular" embolic agent that causes endotheliitis, endothelial denudation, and eventual vessel occlusion), thus allowing for a more complete retropermeation of a bAVM nidus than otherwise possible through an antegrade transarterial route[14]. TRENSH would thus entail bAVM access and retrograde treatment through morphologically simpler DVs, as opposed to the usual more complex AF supply of bAVMs. In turn, this transvenous strategy was hypothesized to likely lessen AF-originating ischemic complications, and allow for larger extents of bAVM nidus occlusion than possible using current transarterial embolization[14]. In essence, TRENSH proposes that the nidus of a bAVM could be successfully retroinjected with a liquid sclerosant (not the currently used viscous agent ethylene vinyl alcohol [EVOH] copolymer dissolved in dimethyl sulfoxide [DMSO] [Onyx®] or its variants, and the polymerizing agent n-butyl cyanoacrylate [NBCA] or its variants) in a hypotensive environment without increasing intranidal pressure and rupture risk[9]. The tenets of TRENSH were subsequently tested in a feasibility and validation study using the pig carotid-jugular fistula-type AVM model[16]—the most widely used animal AVM model[17] since it was first developed in 1994[18]. The use of progressively deeper systemic hypotension levels resulted in lower transnidal pressure gradients across bilateral fast-flowing *retia mirabilia*, enabling transvenous contrast medium injection and angiographic demonstration of retropermeation throughout the simulated nidus. Subsequent in vitro modeling has similarly confirmed this principle by showing that retrograde contrast medium advances into a simulated bAVM nidus in direct proportion to lower mean arterial pressure[19].

To date, the pig AVM model study remains the only in vivo experimental feasibility and validation assessment of TRENSH as a method for retrograde transvenous treatment of bAVMs[16]. That investigation has set the stage for further exploration and refinement of this technique and its underlying principles using similar or improved AVM animal and in vitro models. Moreover, physiologically representative and powerful biomathematical and computational models could provide valuable additional guidance based on greater understanding of underlying hemodynamics[20]. That would be particularly relevant for further analysis of the translocation of therapeutic liquid agents through bAVMs during TRENSH or newer variations on this strategy, all with the aim of increasing the safety and efficacy of future treatments. Overall, there is a pressing need for further critical analysis, refinements, and translational innovations in transvenous bAVM treatment strategies through use of experimental models; this would be key in defining new therapeutic paradigms for the clinical management of patients harboring large bAVMs[3].

In this experimental study we address the specific need to improve the TRENSH technique by application of additional hemodynamic strategies that in theory could render it safer and more effective in treating bAVMs of greater angioarchitectural complexity than possible currently[3]. Accordingly, to improve retropermeation of the nidus during TRENSH we here introduce the broad previously untested concept of implementing hemodynamic manipulations downstream of a bAVM in the venous circulation. To test this concept, we use two different experimental models to simulate mechanistically driven (that is, informed by detailed physiological or physical principles) venous hemodynamic manipulations, and we implement them in conjunction with arterial changes (systemic hypotension and TBO of AFs) already established as necessary during TRENSH[2,9,14]. We use a biomathematical bAVM model incorporating greater nidal complexity than previously available models, as well as the pig *rete mirabile* AVM model[16,18] to investigate the principles and experimental feasibility of enhancing transvenous nidus retropermeation during simulated TRENSH treatments.

## Methods
### AVM hemodynamics
We developed an improved biomathematical, computationally enabled model of brain AVMs (Fig. 1a) based on electrical network analysis to assess the theoretical feasibility of different mechanistically driven hemodynamic maneuvers to enhance TRENSH treatments (Fig. 1b–g). We used this model to simulate TRENSH injections and analyze critical outcomes such as nidus filling and rupture risk. As in previous accounts[21–24], we approximated AVMs as having incompressible, laminar, Newtonian flow through cylindrical vessels of constant and circular cross section, ignoring pressure drops across vessel junctions. We adopted that approximation because blood within most human vessels consists of a non-Newtonian fluid that flows in a pulsatile nature through viscoelastic tapered tubes, which would require much more intensive computational calculations. Moreover, as blood flow in AVMs and surrounding vessels is within thin capillary-like vessels downstream of high capacitance vessels, turbulence, pulsatility, and changes in viscosity are negligible[23]. We therefore suitably approximated AVM hemodynamics flow through each vessel using the Hagen–Poiseuille equation:

$$Q = \pi \Delta P r^4 / 8L\mu$$

Where $Q$ is the volumetric flow rate, $\Delta P$ is the difference in pressure between the ends of the vessel, $r$ is the inner radius of the vessel, $L$ is the length of the vessel, and $\mu$ is the viscosity of blood. We kept blood viscosity $\mu$ at a constant 3.5 centipoise. Vessel lengths and radii were assigned according to Supplementary Table 1, based on prior AVM models[21]. For each vessel, we precalculated a vessel resistance $R_v = (8L\mu)/(\pi r^4)$ so that flow could be calculated as

$$Q = \Delta P / R_v \tag{1}$$

This allowed us to simulate AVM hemodynamics using established methods for modeling electric circuits, with the Hagen–Poiseuille law $Q = \Delta P/R_v$ being analogous to Ohm's law $I = V/R$, where $I$ is the electric current, $V$ is the voltage, and $R$ is the resistance. This analogy allowed simulation of the fluidic network of blood flow within AVM vasculature using established, matrix-based analysis of electrical networks. If pressure, length, and radius are known for all blood vessels, it was possible to calculate blood flow rate throughout the entire AVM.

### AVM electrical network model
In network analysis, vessels of different radii and lengths are randomly distributed in a dependent manner to resemble a highly disordered mesh through which fluid will flow. In analogies within electricity, a circulatory network can be characterized as a complex electrical circuit of connected wires with variable resistance through which current (or flow), powered by an electrical power source (or pressure gradient) will traverse. Each wire (or

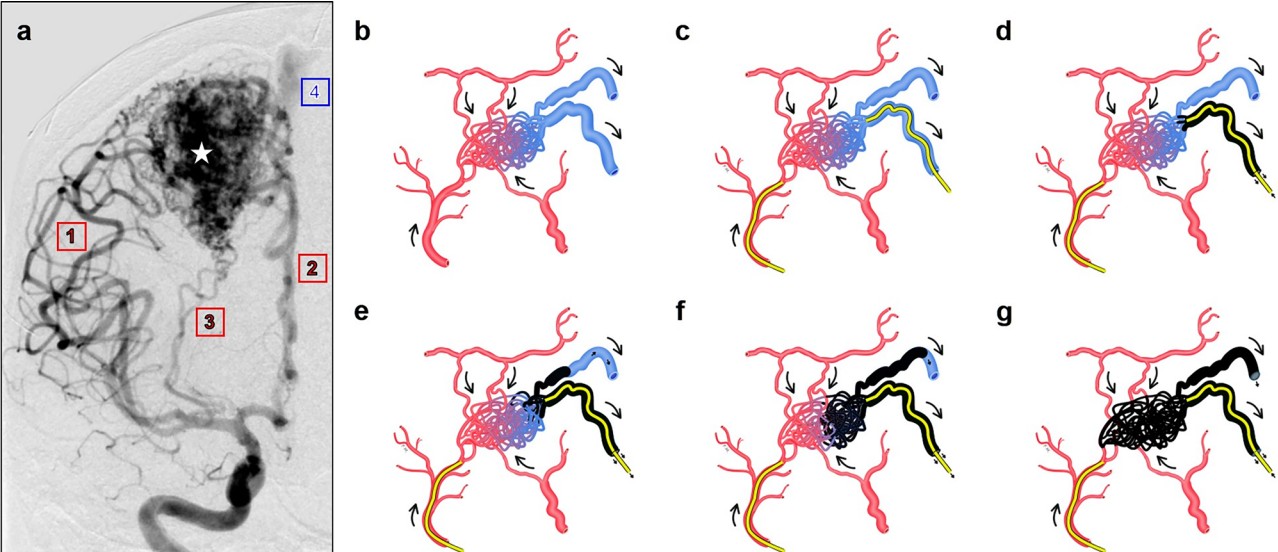

**Fig. 1 | Brain AVMs and principles of TRENSH. a** Frontal right internal carotid angiographic view of a large right frontal lobe AVM nidus (white star) supplied by multiple AFs (red labels: 1. middle cerebral artery branches, 2. anterior cerebral artery branches, and 3. lenticulostriate arteries) and draining through a main DV (blue label: 4. superior sagittal sinus). **b–g** Schematic demonstration of the principles of TRENSH treatment for brain AVMs: **b** Illustration of a hypothetical AVM with multiple AFs in red, two DVs in blue, and a core of tangled microvessels representing the nidus. Arrows indicate direction of flow. **c** Superselective microcatheterization of an AF (for angiography and possible TBO), and of a DV to perform retrograde TRENSH injections. **d** Attempted transvenous treatment at systemic normotension. Sclerosant unable to enter the nidus owing to the strong antegrade flow of blood. **e–g** Progressive phases of TRENSH showing progressive retrograde permeation of the nidus and escape of the sclerosant via the DVs. AFs are not affected.

vessel) connection represents a node (or a vascular intersection) at which flow converges and diverges. With respect to an AVM, a node resembles a vessel bifurcation or trifurcation within the vascular bed. The merits and demerits of electrical network modeling of brain AVMs have been discussed in detail and extensively referenced previously by Hademenos et al.[21] and especially in Jain et al.[23], and these apply here equally (also see a summary of model limitations in the Discussion). Jain et al. achieved a more complex and realistic elaboration on previous theoretical models by introducing two main improvements[23]. They increased the number of intranidal vessels to allow for simulations within anatomically more realistic representations of AVMs, and they introduced extremely high stochastic variability in nidus configurations and complex nidus angioarchitectures, including vessels that connected to more distant nodes.

As in previous accounts[21–24], we modeled an AVM network as a directed graph where each edge represented a vessel, and each node represented a branching point in the vascular network. For the vessels outside the AVM nidus, we used the same architecture (Fig. 2A) and vessel parameters (Supplementary Table 1) as described by Jain et al.[23] This architecture included three DVs (DV1–DV3), two major AFs (AF1 and AF2) with low resistances, and two minor AFs (AF3 and AF4) with high resistances (Supplementary Table 1). We placed a total of nine electromotive forces (EMFs) in the circuit to generate the driving pressures: one (ESP) for the systemic pressure, one for each AF, one for each DV, and one (ECVP) for the CVP. By inputting different values for these EMFs, we were able to simulate different physiological conditions for variations on TRENSH that included testing of: (1) different degrees of controlled systemic arterial hypotension, (2) consequences of TBO of different AFs, (3) different DV retrograde injection pressures, (4) use of different DVs for retrograde injections, (5) elevation of CVP (CVP-high), and (6) retrograde injections via the DVs synchronized to the diastolic phases of the cardiac cycle. The different physiological conditions with their electromotive force values are listed in Supplementary Table 2.

The AVM nidus comprised interconnected plexiform vessels and a single fistulous channel—a larger-radius pathway extending from AF2 to DV2. Earlier theoretical models of AVMs assumed predetermined, fixed intranidal architectures[21,22], which could not capture the substantial structural variability seen across AVMs. To address this limitation, Jain et al. introduced a randomized model in which any two intranidal vessels were equally likely to connect[23]. However, that approach did not reflect the biological reality that spatially proximate vessels are more likely to be connected and that intranidal vessels typically organize into compartments.[14] Our improved model is built upon that of Jain et al.[23] with additional development of a compartment-based algorithm that generates nidus architectures randomly while preserving these spatial and compartmental relationships.

Thus, to generate a nidus, we first set the number of compartments to be a random integer bounded from 3–6 (inclusive), sampled from a normal distribution centered around 4.5 with a standard deviation of 1. We then set the number of columns to be a random integer bounded from 3–7 (inclusive), sampled from a normal distribution centered around 5 with a standard deviation of 1. Each column represented an arbitrary step of distance, such that nodes in adjacent columns were more likely to be connected. Each nidus was divided into horizontal compartments that spanned from AFs on the left to DVs on the right. Columns in the center of the nidus contained a greater number of nodes (about 36–40) than columns near the AFs and DVs (that contained about 30–34). Vessels were then uniformly randomly generated to ensure that each intranidal node had at least one incoming vessel (from the previous column of the same compartment) and one outgoing vessel (to the next column of the same compartment). Each node on the far left/right was connected to the AF/DV node that it was closest to by Euclidean distance. If an AF or DV was left unconnected (this could only occur with AF4), it was connected to the nearest intranidal node.

Grzyska and Fiehler described how even large, multicompartmental AVMs can be entirely filled by a contrast agent from just one injection point and concluded that compartments may be connected by intercompartmental vessels[25]. To reflect this, after generating the intracompartmental vessels, our model then generated [2 × NumCompartments × NumColumns] intercompartmental vessels. Each intercompartmental vessel started on a uniformly random node of a random column of a random compartment. The compartment it ended in was uniformly randomly chosen from the remaining compartments. The column it ended in was sampled from a normal distribution centered around, but excluding, the start column, with a standard deviation of 2. The node it ended on was uniformly randomly

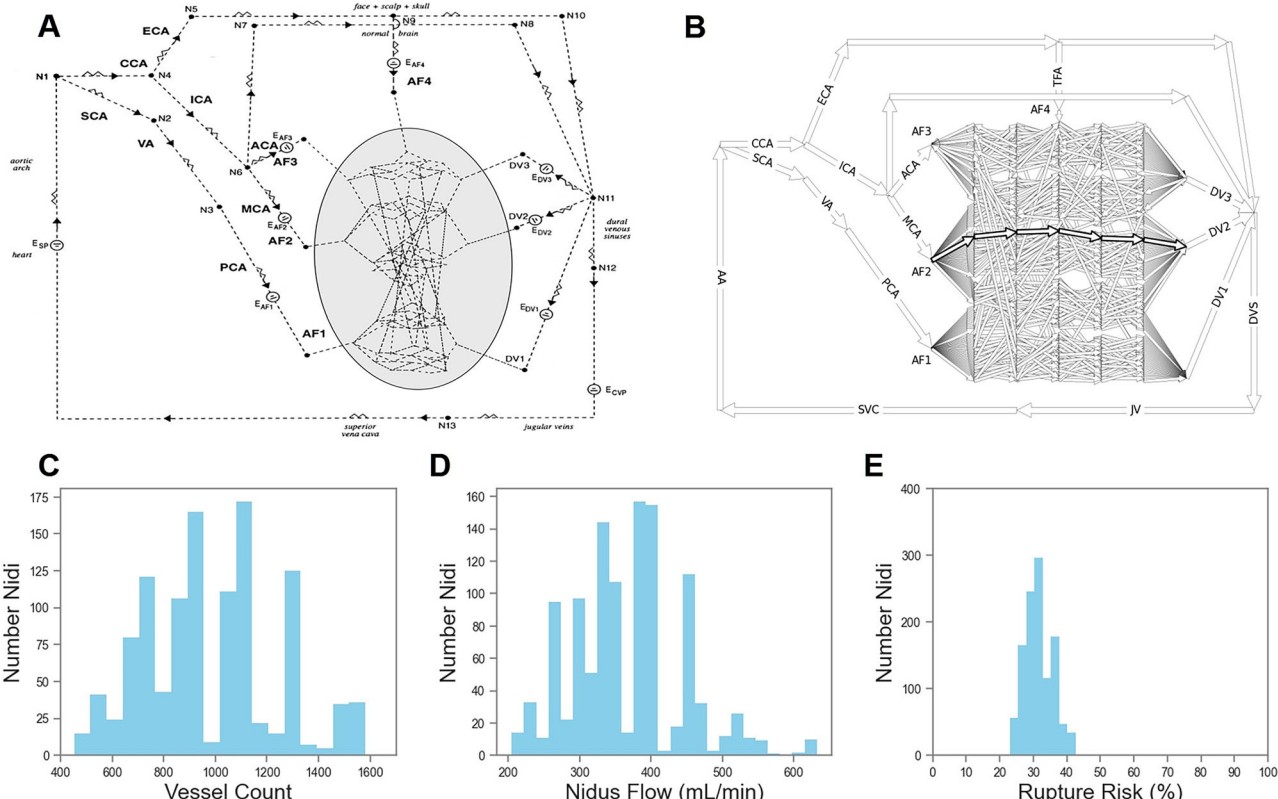

**Fig. 2 | Baseline features of the theoretical AVM model. A** Schematic of the theoretical AVM model. Cluster of vessels within the oval is a stylized representation of multiple interconnected nidus vessels as described in Methods and Results. AF arterial feeder, DV draining vein, CCA common carotid artery, ECA external carotid artery, ICA internal carotid artery, SCA subclavian artery, VA vertebral artery, PCA posterior cerebral artery, ACA anterior cerebral artery, MCA middle cerebral artery, E electromotive force, N node, CVP central venous pressure. Arrowheads indicate direction of flow in the simulations. **B** A schematic version of the theoretical AVM model showing intranidal vessels in greater detail (with five compartments and four columns), and an intranidal fistula spanning AF2 to DV2. AA aortic arch, DVS dural venous sinuses, SVC superior vena cava, TFA transosseous feeding artery. **C–E** Features of the theoretical AVM model in its baseline state. **C** Number of nidus vessels (982 on average) across all architectures. **D** Total intranidal flow (mean of 367 mL/min). **E** Maximum percentage rupture risk among all vessels of the nidus (mean of 31.7%). Distributions drawn from 1139 nidus architectures.

chosen from that compartment and column. Finally, the intranidal fistula started at AF2 and connected to a random node of the first column of the middle compartment. It traveled through a random node of each column of this compartment and ended at DV2.

With this method, we generated 1139 unique nidus architectures. For each architecture, we performed 320 different simulations: 32 with no injection (4 systemic hypotension levels × 2 CVP levels × 4 TBO of AFs states), and 288 with injection (4 hypotension levels × 2 CVP levels × 4 AF TBO states × 3 DV retrograde injection locations × 3 levels of retrograde injection pressures) (see Supplementary Table 3). We used retrograde injections with pressures of 10, 20, and 30 mmHg, which are well within maximal pressure increases upon injections in normal dog arteries[26]. See later for the details of simulations for different strategies to improve on standard TRENSH maneuvers.

**Hemodynamic simulations**

For each simulation, we used its unique vascular architecture plus EMF (pressure) inputs to compute the volumetric flow of each vessel in the network at steady state. Volumetric flow for each vessel was calculated to satisfy Kirchhoff's circuit laws: (1) for every node, the incoming and outgoing flow must be equal; and (2) the sum of the pressure differences of vessels along a closed loop, or cycle, must be zero. The first law follows from incompressibility, and the second ensures that pressure is not lost or gained except from external pressure sources.

From the first law, we had one linear equation for each node: the sum of the (unknown) incoming flows equals the sum of the (unknown) outgoing

flows. From the second law, we had one linear equation for each cycle in the cycle basis of the graph:

$$\sum_i EMF_i = \sum_j \Delta P_j$$
$$= \sum_j R_{vj} Q_j$$

where $EMF_i$ is the $i$-th EMF in the cycle, and $\Delta P_j$ is the pressure gradient of the $j$-th vessel in the cycle. The second line originates from Equation EQ1, where $R_{vj}$ is the vessel's resistance, and $Q_j$ is its flow.

Because these equations together formed a system of linear equations with unknown flow, we could use them to compute vessel flow. Rather than solving with an iterative approach, we leveraged matrix multiplication and vectorized operations to efficiently perform 320 simulations per nidus architecture, yielding a total of 364,480 unique simulations. We used the matrix equation $R_v Q = \Delta P$, where $R_v$ is an m × w matrix of vessel resistances, $Q$ is a w × n matrix of vessel flows, and $\Delta P$ is an m × n matrix of vessel cycle pressure gradients. "m" is the number of equations in the system, "n" is the number of "pressure sets" to simulate, and "w" is the number of vessels. By inputting different "pressure sets"—sets of values for the EMFs used to satisfy Kirchoff's second law—we simulated different physiological conditions to test variations of TRENSH treatments (see Supplementary Table 2). After solving, each column of $Q$ contained the vessel flow rates of one pressure set simulation for the nidus architecture. This method allowed us to simulate for each nidus architecture all 80 pressure sets simultaneously.

We also conducted a limited sensitivity analysis of the model by systematically altering two biophysical parameters (vessel length and radius) to study how variations within a physiologically plausible range might impact model behavior and output fidelity.

All simulations were performed in Python™ version 3.12.0 (Delaware, USA). We used the package NetworkX© to model the vessel architecture as a directed graph, and the package NumPy© to solve the system of linear equations derived from Kirschoff's laws.

## Risk of nidus rupture

It can be reasonably assumed that on the basis of biomechanical properties of the intranidal vessels, rupture occurs when the cumulative hemodynamic stresses of the vessel wall exceed its elastic modulus[27]. Therefore, for each intranidal vessel, we also computed a rupture risk probability using the equation and its derivation described by Hademenos and Massoud[27] and Jain et al.[23]

$$\text{Risk} = \ln\left(\frac{P_{exp}}{P_{min}}\right)\Big/\ln\left(\frac{P_{max}}{P_{min}}\right) \times 100\%$$

where $P_{exp}$ is the pressure of the vessel, which is found by solving for the pressure difference in Equation EQ1, $P_{min}$ is 4 mmHg (based on the minimum CVP in our pressure sets), and 74 mmHg (equivalent to $9.8 \times 104$ dyne/cm$^2$) is the maximum value expected in the AFs during systemic hypertension (as reasoned by Hademenos and Massoud[27]). The expression given in this Equation represents the normalized probability or risk of rupture and is multiplied by 100% to present the results as a percentage of risk of rupture.

## Simulations of TRENSH and its variations

**TRENSH at different systemic arterial hypotension levels and CVPs.** TRENSH was originally proposed by Massoud and Hademenos as an endovascular transvenous therapy for large or complex brain AVMs—an alternative to conventional transarterial embolization[14]. The principle behind TRENSH is to retrogradely inject a sclerosant via a DV of an AVM rather than through an AF, allowing for retrograde transcompartmental permeation of therapeutic sclerosant into the entire nidus. To facilitate this retrograde treatment, the arterial inflow to the AVM would first need to be decreased to allow transvenous nidus filling against the normal antegrade flow. This requires induction of controlled systemic hypotension that affects all visible and occult AF inflow to the nidus and, if necessary, performing TBO of the main AF(s). Here, we tested the addition of CVP-high (from its normal level of 4-6 mmHg to a higher level of 12 mmHg[28]) to aid further in decreasing the transnidal pressure gradient.

Therefore, to assess the efficacy of this TRENSH strategy for each of the 1139 nidus architectures, we simulated endovascular transvenous microcatheter injections of a liquid sclerosant (simplistically assumed to also have a viscosity of 3.5 cP) into each DV at different mean systemic arterial blood pressures (normotension, or mild/moderate/severe hypotension at 74, 70, 50, and 25 mmHg[27], respectively) and CVP levels (normal and elevated[28]). The eight base sets of input pressures ("pressure sets") we used are listed in Supplementary Table 2, with an "intermediate" cardiac cycle phase (meaning mid cardiac cycle; see later for explanation). The pressure sets for different injection pressures through each of the three DVs were then generated by adding 10, 20, or 30 mmHg to the EMF at each of the three DV locations. We used these conservative values for intravascular pressure increases in consideration of the relative fragility of AVM microvessels, compared, for instance, to the near-double values of pressures generated in 1-mm-sized normal internal carotid arteries of dogs by Saitoh et al.[26]

To accurately quantify the extent of retrograde permeation achieved by TRENSH, we developed a graph-theory-based algorithm designed to mirror the physiological characteristics of vascular flow. For each simulation, we first computed the volumetric flow of each vessel at steady state after injection (see earlier, in the "hemodynamic simulations" section), resulting in a directed graph of vessel flow. In this graph, during retrograde injection, a vessel was considered "filled" by the sclerosant only if there existed a continuous intranidal path from the injection site to the vessel's origin node, reflecting the unidirectional nature of blood movement within the vasculature at steady state. This method ensured a physiologically accurate simulation of sclerosant distribution across the AVM's complex structure.

After the retrograde injection was completed, the hemodynamic conditions within the AVM shifted as the sclerosant exited the nidus antegradely (owing to the normal inflowing arterial blood), resulting in a different flow profile. To capture this change, we extended our algorithm to model post-injection nidus emptying by recalculating the steady-state blood flow using the directed graph generated from the baseline (no-injection) pressure set. In this post-injection state, a vessel was considered "filled" if it could be reached along an intranidal path from any vessel that was already filled during the injection phase. By modeling both the entry and exit phases of the sclerosant, we ensured a comprehensive and physiologically accurate representation of nidus permeation throughout retrograde treatment of the AVM.

**TRENSH with temporary balloon occlusion of arterial feeders.** Disruption of intranidal hemodynamic balance after TBO of one AF at a time has been demonstrated in clinical settings previously[29]. It is reasonable to assume, therefore, that a retrograde injection via a DV during TRENSH would also alter the intranidal hemodynamic equilibrium (especially in the presence of intranidal hypotension) to allow retrograde spread of sclerosant throughout the nidus. TBO during transvenous embolization of bAVMs has been reported by Iosif et al.[30]

To assess whether and to what extent TBO of AFs could facilitate retrograde nidus permeation during TRENSH, we modeled the effects of theoretical AF occlusion on the sclerosant distribution within the AVM nidus. For each of the 1139 nidus architectures, we repeated the DV retrograde injections with either AF1, AF2, or AF3 occluded by removing the corresponding edge from the graph, effectively blocking any flow through that vessel. We also investigated if occlusion could enable TRENSH intranidal retropermeation to occur at lower DV retrograde injection pressures. To perform a more detailed analysis, we additionally generated a separate set of 61 nidus architectures and repeated the TBO simulations with injection pressures ranging from 20 to 30 mmHg in increments of 1 mmHg.

**TRENSH using synchronized diastolic retroinjection and nidus permeation.** We conducted simulations to determine if the timing of TRENSH retrograde injection relative to the cardiac cycle might influence effectiveness in retrograde permeation of the nidus. Our conceived adaptation of this for use in bAVMs was analogous in principle to the strategy attempted in the past for electrocardiogram-triggered synchronized diastolic retroperfusion of ischemic heart myocardium[31,32]. More specifically, we aimed to assess whether, in theory, the synchronization of TRENSH retrograde injections during the diastolic phase of the inflowing arterial blood flow (when intranidal blood pressure was at its lowest) would result in more effective retrograde nidus permeation (because the nidus would be relatively more flaccid in its diastolic phase) compared to the systolic phase when intranidal pressure is at highest. For this analysis, we generated 61 nidus architectures, and for each one we performed 1360 simulations (the variables being three "phases" × 12 injection pressures), all under mild systemic arterial hypotension, normal CVP, and with retrograde injections through DV1. The "phases" in this context were the three cardiac cycle phases of "diastole", "intermediate/mid cycle, or mean", and "systole".

**Animation of simulated TRENSH treatments.** To visualize and better understand the sequential progressions of each of our tested innovative strategies for TRENSH simulations in our theoretical bAVM model, we generated animations of nidal filling by arbitrarily coloring blue the circuit vessels that a sclerosant reached as determined by our biomathematical calculations. Each animation had two phases that reflected

permeation of nidus vessels by transvenous treatment through the DVs: a retrograde ingress phase of simulated sclerosant during injection, and an antegrade exit phase of sclerosant upon stopping the injection. The retrograde ingress phase consisted of one frame for each injection pressure from 0 mmHg up to the target injection pressure in increments of 1 mmHg. In these frames, vessels were colored blue if there existed a path in the directed graph from the injection node to the vessel's entry node. Upon dropping the injection pressure back down to 0 mmHg to signify the end of the injection, the antegrade exit phase consisted of the paths of the sclerosant as it exited the nidus. For each of these frames, a vessel was colored blue if and only if a vessel pointing to its entry node was colored blue in the previous frame. Each simulation produced a set of PNG images that visualized the filling of vessels frame by frame. We stitched together these images into an MP4 video using the Python[TM] package imageIO 2.37.0 to provide a comprehensive view of the simulated intranidal sclerosant dynamics.

**Statistics and reproducibility.** We conducted statistical analysis using unpaired two-tailed Student's $t$-tests to compare flow, nidus filling, and rupture risk across experimental conditions. Each test compared two pressure sets that differed only in a single parameter of interest (e.g., systemic blood pressure, CVP, injection pressure, DV selection, or the use of TBO), while all other parameters were held constant. The specific pressure sets compared are described in the Results. We calculated group means from repeated simulations within each condition. Independent replicates were defined as distinct randomly generated nidus architectures, with $n = 1139$ architectures used for the main analyses and $n = 61$ for the sub-analyses; outputs were averaged across all architectures contributing to each condition. Simulations were deterministic for a given architecture and pressure set, ensuring reproducibility. A $P$-value of less than 0.05 was considered statistically significant. We performed all analyses using Python (v3.12) and the SciPy library.

**Simulations of TRENSH and its variations in an animal model**
**Pig AVM model.** We prospectively conducted all animal experiments in accordance with policies set by UCLA's University Chancellor's Animal Research Committee (approved protocol 1998-013) and National Institutes of Health guidelines. We used six Red Duroc pigs; animals were three to 4 months old, weighed 30–40 kg, of mixed sex, and maintained on a standard laboratory diet. After an overnight fast, we premedicated each pig with intramuscular 20 mg/kg ketamine and 2 mg/kg xylazine, and maintained general anesthesia with mechanical ventilation and inhalation of 1–2% halothane after endotracheal intubation.

The pig carotid rete mirabile is a network of micro-arteries (average diameter 154 μm)[33] situated at the end of each ascending pharyngeal artery as it penetrates the skull base. It has morphological similarities to a human plexiform AVM nidus but lacks the additional hallmark of arteriovenous shunting. We introduced this requisite rapid shunting by surgical construction of a carotid-jugular fistula to produce the pig AVM model, comprising a simulated nidus made of bilateral retia mirabilia and their inter-retial connections (see "Results"). Details of model construction, its baseline angiographic and hemodynamic features, and its advantages and limitations have been extensively studied and discussed in previous publications[16,18]. At the termination of each study, pigs were euthanized under deep anesthesia with an overdose of a barbiturate solution, consistent with AVMA and institutional guidelines.

**TRENSH at different systemic arterial hypotension and venous hypertension levels.** We tested TRENSH treatments in the pig AVM model at different levels of arterial hypotension and venous hypertension; the latter hemodynamic maneuver to imitate the raising of "CVP" to aid nidus retropermeation during simulated TRENSH. First, we obtained a selective left common carotid angiogram to reveal the global vascular components of the AVM model. Next, after navigating and positioning two 2.2F microcatheters in bilateral ascending pharyngeal arteries, we hemodynamically assessed each AVM through its main AF and DV at systemic normotension (syst-Normo). We obtained mean transcatheter pressure recordings in the AF (MAP) and DV (MVP), and then performed both transarterial and retrograde transvenous nidus superselective angiography at baseline syst-Normo in all pigs. In addition, in three of the six pigs, superselective antegrade and retrograde angiograms of the nidus were obtained about 2 min after each 75 mL blood aliquot was drawn from the pig (to induce hemorrhagic systemic hypotension [syst-Hypo], see later). For consistency, the same operator performed all angiograms by using 0.8 mL of non-ionic contrast medium hand injected with a 1 mL syringe over a 2 s period.

In previous studies, TRENSH was simulated in pigs by transvenous retrograde permeation of the nidus using contrast medium, and angiographic documentation of its spread at progressively deeper levels of syst-Hypo[16]. In this hemodynamic study, we similarly used contrast medium because the aim was as to repeatedly document at angiography the retrograde spread of an injected liquid (contrast medium in this case, and not a sclerosant per se) within the nidus, and to correlate the extent of this with the degree of reduced transnidal pressure gradient brought about by manipulations of arterial and venous pressures across the AVM nidus.

Next, we qualitatively graded the extent of transvenous contrast medium retropermeation of the nidus, as described previously[16], from the least amount (Stage 1: minimal nidus penetration by contrast medium and simultaneous emptying of any observable nidus filling through the main DV around the microcatheter used for injection), to the greatest amount (Stage 6: retrograde filling of most of the nidus, including if there is further undesirable spill-over filling of the main AF). We then correlated each angiographic image and corresponding "Stage" of retrograde permeation with the arteriovenous pressure gradient across the nidus and the degree of syst-Hypo and venous hypertension present.

We qualitatively graded nidus retropermeation upon simulating TRENSH in two groups of three pigs each. In Group 1 pigs, we did not induce syst-Hypo. Instead, at syst-Normo, we tested the hypothesis that introducing temporary venous hypertension alone, originating considerably distal to the DV itself (by either abdominal compression for 15 s to raise "CVP" in two pigs, or partially clipping the internal jugular vein beyond the carotid-jugular fistula in one pig), would be transmitted to the DV, to then allow enhanced retropermeation of the AVM nidus upon retrograde injection of contrast medium to simulate a TRENSH treatment. Of note, the transvenous angiograms were performed at the end of the abdominal compression period or during DV clipping. In Group 2 pigs, the same hypothesis was tested but with the addition of syst-Hypo. Thus, TRENSH was simulated in two pigs at syst-Hypo plus partial DV clipping, and in one pig at syst-Hypo plus abdominal compression.

We induced hemorrhagic syst-Hypo in these Group 2 pigs by repeated withdrawals of 75 mL aliquots of arterial blood through the right common carotid artery guiding catheter. The total amount of blood removed from each animal was about 1 L over 1 h. For the purpose of this study, the MAP was considered to be proportional to the mean systemic blood pressure (as shown by Massoud previously[16]). Consequently, in our experiments, it was not necessary to simultaneously record systemic blood pressure in a more central body location. Thus, the recorded reduction in MAP was deemed to be a direct reflection on the drop in systemic pressure consequent to bloodletting.

## Results
### Baseline biomathematical AVM model features
We first validated the theoretical physiological accuracy of our biomathematical model (Fig. 2A, B) at baseline conditions, that is, at systemic normotension and no simulated TRENSH retrograde DV injections. Across all 1139 nidus architectures, the average vessel count was 982 vessels (range 453–1769) (Fig. 2C). The average total volumetric flow through the nidus was 367 mL/min (range 205–632 mL/min) (Fig. 2D), approximating the average flow of 358 mL/min described in bAVMs previously[34]. The calculated rupture risk averaged 31.7% (range 23.0–42.6%) (Fig. 2E). For a limited

sensitivity analysis of this model, we generated 1000 random nidus architectures with vessel length and radius sampled uniformly within ±10% of their typical values listed in Supplementary Table 1; this resulted in volumetric blood flow through the AVM nidus ranging from 162 to 796 mL/min, which aligns with expected physiological variability[34] and confirms the robustness of the model to small variations in key vessel parameters.

### Effects of systemic arterial hypotension, additional CVP elevation, and transvenous injection pressures on intranidal flow during TRENSH

It was previously proposed that systemic hypotension could facilitate TRENSH by decreasing a bAVM transnidal pressure gradient[14]. In the present study, we build on that previous work by testing if additional temporary central venous pressure elevation (CVP-high)—achievable in clinical settings using positive end-expiratory pressure ventilation during anesthesia[28]—would result in further diminishing of transnidal flow and pressure gradient to allow greater nidus retropermeation during TRENSH. To verify this theoretically, we first simulated the baseline average total volumetric nidus flow at different levels of systemic hypotension (defined in "Methods") with and without CVP-high (Fig. 3A), both before and after simulated retrograde injections at different injection pressures varying from 10 mmHg to 30 mmHg (Fig. 3B)[26]. Before injection, we found that at a normal CVP (CVP-norm) of 6 mmHg, the average total flow dropped from 367 mL/min at systemic normotension to 105 mL/min at profound hypotension (P < 0.001) (Fig. 3B). We observed that with CVP-high at 12 mmHg, the average total flow dropped from 326 mL/min at systemic normotension to 94 mL/min at profound hypotension (P < 0.001) (Fig. 3B). With simulated retrograde injections during normotension and all other tested levels of systemic hypotension, the additional maneuver of raising CVP similarly resulted in significant diminished intranidal total volumetric flow (P < 0.001) (Fig. 3B). Moreover, the stronger the simulated retrograde injections of sclerosant, the lesser antegrade flow occurred through the nidus.

### Effects of systemic arterial hypotension, additional CVP elevation, and transvenous injection pressures on extent of intranidal retropermeation during TRENSH

To quantify the theoretical effectiveness of simulated TRENSH, we computed nidus filling upon retrograde injection. Retrograde filling was highly dependent on the degree of systemic hypotension and injection pressure through the DV (Fig. 3C). At normotension with CVP-norm, the average filling achieved with a 30-mmHg injection was 4.0%; at minor systemic hypotension it was 4.5%; and at moderate systemic hypotension it was 16.2%. Only at profound systemic hypotension did TRENSH achieve an average nidus filling of 87.4%, which was significantly higher than at any other systemic blood pressure condition (P < 0.001 for each comparison) (Fig. 3C). Simulated CVP-norm versus CVP-high had a small but significant impact on the effectiveness of retrograde nidus filling during TRENSH. Thus, at profound hypotension with a 30-mmHg injection, CVP-high increased retrograde nidus filling from 87.4 to 88.7% but this was statistically significant (P < 0.001). At profound hypotension with a 20-mmHg injection, CVP-high also increased filling from 33.0 to 37.0% (P < 0.001) (Fig. 3C).

### Effects of systemic arterial hypotension, additional CVP elevation, and transvenous injection pressures on mean risk of nidus rupture during TRENSH

To evaluate the theoretical safety of TRENSH, we calculated the mean nidus rupture risk for each TRENSH simulation. Rupture risk substantially decreased with greater systemic hypotension owing to the lower transnidal flow and pressure gradient, which likely reduced intravascular stress. We observed that lowering mean systemic blood pressure from moderate to profound hypotension resulted in the greatest benefit in lowering rupture risk; with a 30-mmHg injection at CVP-norm, the mean rupture risk decreased from 16.7 to 2.8% (P < 0.001) (Fig. 3D). CVP-high contributed

further to lowering mean rupture risk from 2.8 to 1.9% (P < 0.001) (Fig. 3D). For maximal rupture risk variations, see Supplementary Fig. 1.

### Effects of retrograde injection through different draining veins at different injection pressures, with and without CVP elevation, on intranidal retropermeation during TRENSH

To determine theoretically if the injection location influences the effectiveness of TRENSH, we simulated separate injections into three different DVs (Fig. 3E). The average filling for a 30-mmHg injection at profound hypotension and CVP-norm into DV1, DV2, and DV3 was 88.2%, 99.2%, and 74.8%, respectively (ANOVA, P < 0.001) (Fig. 3E). Retrograde injection through DV2 produced the greatest intranidal filling (see Supplementary Movie 1). We surmised this was owing to a direct pathway between DV2 and AF2 via the simulated intranidal fistula; AF2 had the highest flow among the AFs, generating high pressure in its corresponding compartment. As a result of this high pressure, a DV2 injection allowed the simulated sclerosant to flow easily into surrounding lower-pressure compartments, leading to more comprehensive nidus filling[22]. Conversely, DV3 was opposite AF3, which had the lowest flow, resulting in poor filling as the sclerosant became confined to a low-pressure area and spread less to other compartments (see Supplementary Fig. 2, and Supplementary Movies 2 and 3). We then tested the effects of CVP-high on these simulated TRENSH treatments through different DVs and at different injection pressures (Fig. 3F). As shown earlier, CVP-high resulted in a small but statistically significant (P < 0.001) increase in retrograde nidus filling during simulated TRENSH (Fig. 3F).

### Effects of systemic arterial hypotension and different retrograde injection pressures, with additional TBO of arterial feeders, on intranidal retropermeation during TRENSH

To assess whether simulated TBO of AFs could enhance TRENSH effectiveness and potentially allow the procedure to be performed with lower injection pressures through DVs, we simulated TRENSH with sequential TBO of each AF (Fig. 3G–I). At profound hypotension with a 20-mmHg injection, occluding AF2 increased the nidus filling from 33.0 to 70.0% (P < 0.001) (Fig. 3G). This improvement, although substantial, still fell short of the 87.4% filling achieved with a 30-mmHg injection without any TBO, suggesting that while feeder occlusion can enhance filling, higher transvenous injection pressures may still be necessary to achieve the best nidus retropermeation. To gain a more granular understanding of how TBO could facilitate TRENSH at moderate and profound systemic hypotension levels, we analyzed the effects of occluding different AFs across a range of DV injection pressures from 20 to 30 mmHg (Fig. 3J). Under moderate hypotension, occluding AF2 consistently resulted in the highest nidus filling across all pressures, followed by AF1, AF3, and no TBO (P < 0.05). At profound hypotension, a similar pattern was observed until the injection pressure reached 28 mmHg, at which point AF1 occlusion reached 95.7% nidus filling, surpassing the 90.1% filling from AF2 occlusion (P < 0.001) (Fig. 3J). One notable difference at profound hypotension was that the values for AF3 occlusion and no occlusion were not significantly different (P > 0.05).

### Effects of synchronized diastolic retroinjection on intranidal retropermeation during TRENSH

Theoretically, we also observed that the phase of the cardiac cycle during which we performed the simulated injections had mostly—but not uniformly—a minimal effect on nidus filling (Fig. 4). This is likely because of the baseline very small pulse pressure (i.e., a small/shallow difference between peak systolic and trough diastolic pressures) within nidus microvessels (Fig. 4a)[35]. This scenario is unlike past attempts at applying synchronized diastolic retroperfusion to ischemic myocardium in the presence of a much wider pulse pressure conducive to a selective injection during a deep diastolic pressure trough. At systemic moderate hypotension, there was no significant difference between the filling achieved by injections synchronized with any of the cardiac cycle phases (P > 0.05) (Fig. 4b). At systemic

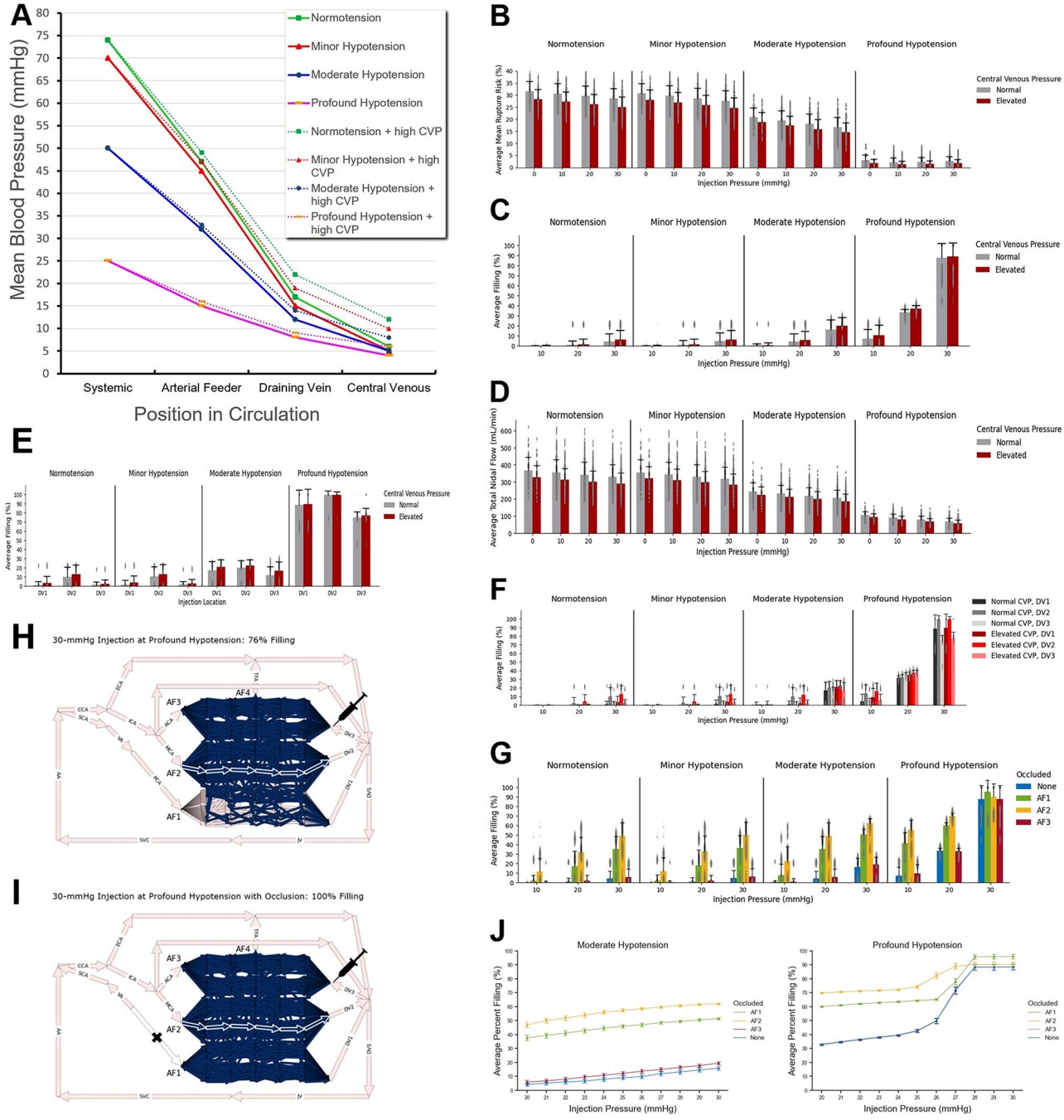

**Fig. 3 | Hemodynamic simulations of TRENSH using the theoretical AVM model.** **A** Line graph representing mean blood pressure drops across the theoretical AVM model according to position within the circulation. Solid lines are in presence of normal CVP; and dotted lines are in presence of elevated CVP. At normotension, mean systemic pressure is 74 mmHg. Systemic hypotension was simulated at 70 mmHg (for minor hypotension), 50 mmHg (for moderate hypotension), and 25 mmHg (for profound hypotension). See Supplementary Table 2 for changes in these pressures across the AVM upon simulation of additional CVP elevation. **B** Bar chart of the average total nidal flow without AF occlusion across all 1139 architectures, grouped by injection pressures, hypotension state, and CVP ($n = 3417$ simulations per bar). **C** Bar chart of the average nidus filling, grouped by hypotension state, injection pressures, and CVP ($n = 3417$ simulations per bar). **D** Bar chart of the average mean rupture risk, grouped by hypotension state, injection pressures, and CVP ($n = 3417$ simulations per bar). **E** Bar chart of the

average nidus filling with a 30-mmHg injection, grouped by hypotension state, injection location in the DVs, and CVP ($n = 3417$ simulations per bar). **F** Bar chart of the average nidus filling, grouped by hypotension state, injection pressures, CVP, and injection location ($n = 1139$ simulations per bar). **G** Bar chart of the average nidus filling with normal CVP, grouped by hypotension state, injection pressures, and AF occlusion location (injection locations are aggregated, $n = 3417$ simulations per bar). For **B**–**G** error bars represent standard deviation. Schematic diagrams showing an example of a 30-mmHg injection into a nidus through DV3, at profound systemic hypotension and with no AF occlusion (**H**, 76% nidus filling) versus with AF1 occlusion (**I**, 100% nidus filling). **J** Graphs showing average percent filling at normal CVP as a function of injection pressure incrementally increased from 20 mmHg to 30 mmHg, and grouped by hypotension state (moderate or profound) and AF occlusion location (injection pressures are aggregated, $n = 183$ simulations per point). For **J** error bars represent 95% confidence intervals.

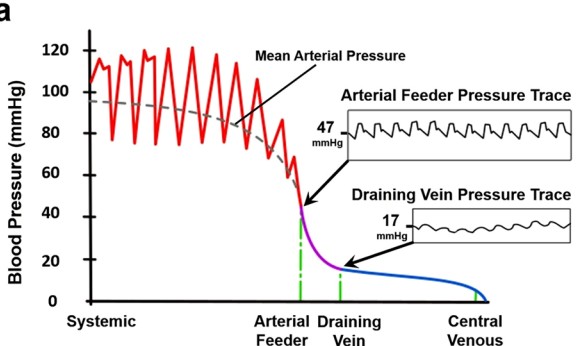

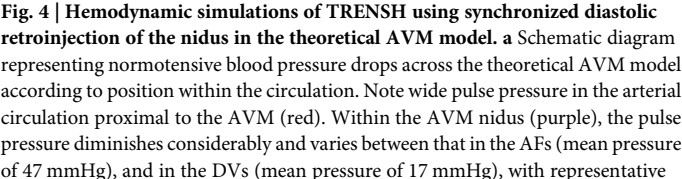

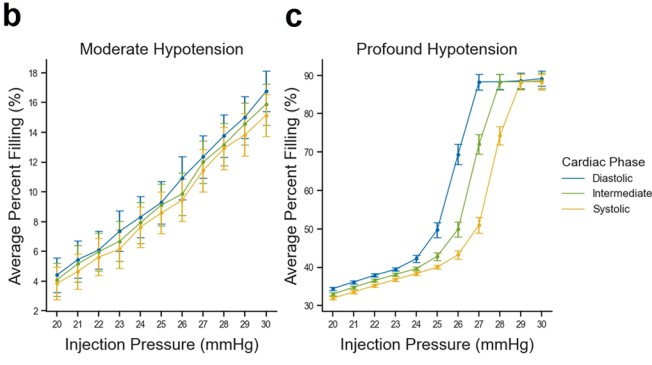

**Fig. 4 | Hemodynamic simulations of TRENSH using synchronized diastolic retroinjection of the nidus in the theoretical AVM model. a** Schematic diagram representing normotensive blood pressure drops across the theoretical AVM model according to position within the circulation. Note wide pulse pressure in the arterial circulation proximal to the AVM (red). Within the AVM nidus (purple), the pulse pressure diminishes considerably and varies between that in the AFs (mean pressure of 47 mmHg), and in the DVs (mean pressure of 17 mmHg), with representative pressure traces for AF and DV reproduced from Nornes and Grip[35]. Venous pressures drop further distal to the AVM nidus (blue). Graphs showing average percent filling across 61 architectures at normal CVP as a function of injection pressure incrementally increased from 20 mmHg to 30 mmHg, and grouped by systemic hypotension state (**b** moderate, and **c** profound) and cardiac phase (injection pressures are aggregated, $n = 183$ simulations per point). Error bars represent 95% confidence intervals.

profound hypotension, there was also no significant difference between the filling achieved by 30-mmHg injections synchronized with the cardiac phases ($P > 0.05$). However, at lower injection pressures near 26 mmHg, we observed a pattern where the diastolic phase achieved the highest filling, followed by the intermediate/mean phase, and then the systolic phase ($P < 0.001$) (Fig. 4c).

### TRENSH testing with and without simulated raised CVP in the pig model

Next, for animal model studies, all pigs tolerated the general anesthesia and surgical and endovascular procedures with no ill effects. We successfully used the pig *rete mirabile* (Fig. 5A) to construct all AVM models, which resulted in a clear baseline transarterial angiographic demonstration of fast shunting through the simulated main "AF" (left ascending pharyngeal artery), the 'nidus' (bilateral retia mirabilia), and the main "DV" (right ascending pharyngeal artery and right common carotid artery down to the carotid-jugular fistula) (Fig. 5B–E, and Supplementary Movie 4)[16,18,36].

We performed all TRENSH simulations and hemodynamic studies on AVM models immediately after their creation. At baseline systemic normotension (syst-Normo), the mean arterial pressure (MAP) and mean venous pressure (MVP) values for all six pigs was 70.6 mmHg and 47.0 mmHg, respectively (akin to previous reports[16]), resulting in a mean transnidal pressure reduction (MVP/MAP × 100%) of 66.5%. For qualitative grading of the extent ("Stage") of transvenous retrograde permeation of the nidus upon transvenous injection through the DV, see "Methods" and Fig. 6A. In all pigs there was minimal retrograde nidus filling (Stage 1 or 2) upon superselective transvenous angiography via the DV at baseline syst-Normo.

In Group 1 pigs, that is, those tested at syst-Normo alone, there was almost no benefit in raising the venous pressure for the purpose of achieving greater retropermeation of the nidus in simulated TRENSH injections, as also predicted in the computational model results shown in Fig. 2. In experimental animals, raised CVP can be achieved by various methods: PEEP, abdominal compression, neck compression, intrajugular balloon inflation, and hypervolemia[37]. In both pigs subjected to abdominal compression (as used by Wauters et al.[38]), there was only a minimal improvement in retropermeation of contrast medium, from a Stage 1 to a Stage 2 grade (see Supplementary Fig. 3 for angiograms of pig #1, and Fig. 6B1–B3 for angiograms of pig #2). Further, the initially observed Stage 2 retropermeation remained unchanged for pig #3 (see Supplementary Fig. 4) receiving partial clipping of the venous outlet.

In Group 2 pigs, that is, those tested initially under syst-Normo and then with additional systemic hypotension (syst-Hypo), there was an overall benefit when temporary venous hypertension maneuvers (that simulate CVP-high) were performed in the presence of additional induction of syst-Hypo. During induction of syst-Hypo we recorded a gradual reduction in MAP in all three pigs over the 1 h period of deliberate exsanguination (see Massoud for methods to produce systemic hypotension in experimental animals[16]). A previous report had indicated that such hemorrhagic syst-Hypo, manifesting in progressive AF hypotension, produces a strong linear correlation between the drop in MAP and MVP across the nidus[16].

Transvenous angiograms in Group 2 pigs revealed progressive increases in Stages of nidus retropermeation when deeper syst-Hypo was augmented by temporary venous hypertension, that is, the two combined strategies produced a greater reduction in transnidal pressure gradients. This is exemplified by pig #4 (Fig. 6C1–C8) and #5 (Supplementary Fig. 5) where venous clipping alone caused minimal increased nidus retropermeation, but sequential clipping maneuvers worked in concert with progressive syst-Hypo to achieve near complete retrograde filling of the nidus. Similarly, for pig #6 (Supplementary Fig. 6), successive 15 s abdominal compression maneuvers also enhanced the effects of progressive syst-Hypo in achieving significant retropermeation of the nidus when simulating TRENSH.

## Discussion

The role of transarterial endovascular embolization is mostly to reduce nidus volume as an adjunct to microsurgery or radiosurgery. Monotherapeutic transarterial embolization is possible in specific clinical situations but is uncommon[39], for instance in small bAVMs with one to three superficial AFs[8], when comorbidities preclude surgical treatment[40], and for palliation in higher Spetzler-Martin grade bAVMs[8,40]. The rates of endovascular cure for brain bAVMs using this method alone vary from <40[3] to 51%[41]. Importantly, bAVMs with *en passage* feeders supplying normal brain parenchyma have low angiographic occlusion rates and higher risks of complications with transarterial embolization[8]. Overall, significant neurological complications leading to disability occur in 11% of endovascular monotherapy cases, largely owing to hemorrhagic events[42]. Efforts to refine endovascular techniques by deeper understanding of pre- and post-therapeutic hemodynamics[43], the use of newer embolic agents and catheter-based tools/devices, as well as innovative treatment strategies may reduce associated complication rates. Several grading systems exist to predict the risk of complications related to endovascular AVM embolization[44].

The transvenous approach to AVMs is considered as a novel and elegant approach to treat a complex cerebrovascular disease[9]. TRENSH was originally conceived in great part with the lofty goal of creating an endovascular treatment (perhaps an attempted cure) for a wide selection of bAVMs that could span all sizes, grades, and angioarchitectural complexities[14]. Please refer to the two prior accounts that discuss in detail the

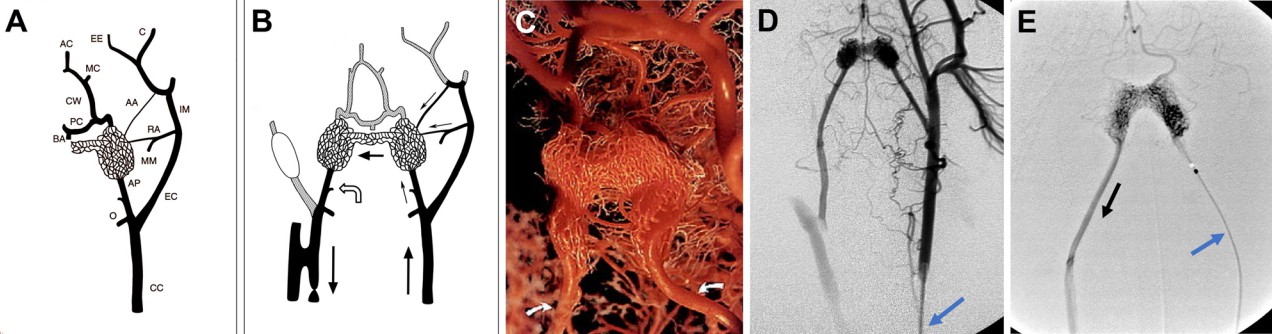

**Fig. 5 | Pig AVM model. A** Schematic of the normal left carotid arterial anatomy of the pig head and neck, reproduced from Massoud et al.[18] CC common carotid artery, EC external carotid artery, IM internal maxillary artery, MM middle meningeal artery, RA ramus anastomoticus, AA arteria anastomotica, AP ascending pharyngeal artery, O occipital artery, BA basilar artery, CW circle of Willis, PC posterior cerebral artery, MC middle cerebral artery, AC anterior cerebral artery, B buccal artery, P palatine artery, IO infraorbital artery. The carotid *rete mirabile* is situated at the termination of AP. In the pig, the internal carotid artery is very short and joins the rete to CW. **B** Schematic of the AVM model as seen on angiography after creation of a right carotid-jugular fistula. See also Supplementary Movie 4. Some features of the original AVM model are depicted for consistency with previous accounts, such as balloon occlusion of the right EC. It was subsequently established that this maneuver, whether performed or not, did not significantly affect the degree of sump effect at the fistula and the accompanying shunting across the *retia* (referenced in Massoud[16]). Ligation of the right CC below the fistula was also an original feature but not performed in this study to allow placement of a transarterial guiding catheter in situ. Straight arrows indicate direction of flow diversion from the left to the right side via bilateral *retia mirabilia* (the "nidus"). Curved arrow indicates muscular branch of AP in which flow reverses owing to the sump effect of the carotid-jugular fistula. **C** Detail from a plastic cast of bilateral carotid *retia mirabilia* of the swine shows the complex branching pattern of this microvascular bed. The surrounding nonvascular structures have been removed. Arrows indicate left and right ascending pharyngeal arteries. Reproduced from Massoud et al.[36] Selective (**D** with catheter in the left CC [blue arrow]) and superselective (**E** with the microcatheter in the left AP [blue arrow]) angiograms of the AVM nidus. Arrow in (**E**) indicates direction of flow.

principles and rationale for this treatment[14,16]. In short, TRENSH is predicated on having in place certain prerequisites: First, the bAVM should be amenable to safe retrograde microcatheterization through DV(s) to closely reach the nidus. Second, advancements in endovascular flexible microcatheters and tools to allow such safe navigation and trackability up to a bAVM nidus on both arterial and venous sides (to also place pliable AF balloon microcatheters). Third, collaborations with colleagues in anesthesia and interventional cardiology to achieve safe temporary levels of profound systemic hypotension and/or cardiac standstill to lower considerably or eliminate the transnidal pressure gradient and to allow safe nidus retropermeation[2,45–47]. This would also disrupt the functional/hemodynamic intranidal compartmentalization of blood flow to allow retropermeation of most or all of a large nidus. Moreover, partial devitalization of the nidus through conventional transarterial embolization could act in concert with TRENSH to reduce its size and intranidal pressure—a dual retrograde and antegrade strike on the nidus, especially when large.

For the fourth prerequisite, the "embolic" agent needs to be a liquid, with low viscosity. This would allow it to easily permeate the entire nidus upon disruption of any pre-existing normotensive compartmentalization, and crucially, to easily escape via the DVs (the same one used for microcatheter retroinjection plus any other DVs) so as to decompress the nidus and avert a pressure buildup owing to the injection pressure. In that respect, a multiplicity of DVs is an advantage in the TRENSH technique when using a liquid. Even though a polymerizing agent is generally contraindicated in current transvenous techniques, the use of viscous, gel-like, non-adhesive embolic agents may still lead to the possibility of solidification within DVs (visible and occult ones) when parts of the nidus with afferent inflow may still be patent, as stated earlier. In the absence of prolonged, and therefore impractical, circulatory arrest, any such venous outlet obstruction for any length of time might theoretically be sufficient to rupture a bAVM nidus that still receives an antegrade arterial supply. Therefore, by using a liquid agent (that ought to have no or inconsequential adverse effects when it disperses beyond a bAVM nidus), the TRENSH technique would also respect the DVs just as in conventional arterial embolization and during microsurgical resection of bAVMs. It could lead to a more controlled and precise nidus treatment, while avoiding immediate and premature occlusion of DVs. In reality, the current scenario of using transvenous treatments solely in small

bAVMs (with a small nidus and a single DV) is akin to treating a single-hole arteriovenous fistula where occlusion of the DV and an adjacent portion of the nidus can disrupt the pressure head and thrombose the rest of the bAVM[9]. Clearly, this strategy would be suboptimal for most AVMs that are large and with complex angioarchitecture.

Finally, a liquid injectate should be a sclerosant that chemically or molecularly damages endothelial cells of AVM nidus microvessels selectively without any deeper damage that might possibly result in vessel rupture. The ensuing endotheliitis and endothelial necrosis or apoptosis would lead to denudation and subsequent surface thrombus buildup leading to eventual sclerosis and occlusion of nidus microvessels over days to weeks. We have yet to establish a sclerosant with these properties, although several candidates have been proposed previously[14].

The principles of TRENSH have laid the foundation for all subsequent clinical studies using the endovascular transvenous approach to treat small bAVMs[10], starting with the first clinical application by Nguyen et al. in 2010[48]. Indeed, in recent years transvenous embolization has emerged as an alternative, potentially curative option for bAVMs, with current indications for this approach being a small (<3 cm) compact nidus, deep bAVM location, hemorrhagic presentation, presence of a single DV, inaccessible AFs, exclusive arterial supply by perforators or choroidal arteries, *en passage* AFs, and absence of a good surgical option or failed radiosurgery[2,9,49]. Adenosine or rapid ventricular pacing-induced systemic hypotension[2,45–47] and TBO of AFs[2,9] when possible must be considered to establish transnidal pressure gradient reduction and flow arrest, along with use of Onyx or another non-polymerizing, non-adhesive embolic agent. A 2019 systematic review and meta-analysis included 66 cases of bAVMs transvenously treated using Onyx, and reported a technical complication rate of 8%, with an overall good functional outcome in 89% of patients[13]. Even though the transvenous approach to bAVMs forgoes the principle of arterial devascularization used in microsurgical and endovascular transarterial bAVM treatments, it aims to achieve the same end result—a successful transvenous embolization still eliminates the nidus before potential venous outflow obstruction occurs[9].

Unfortunately, bAVMs with large (>3 cm) nidi and multiple DVs have revealed a statistically significant propensity for hemorrhagic complications when treated transvenously[50]. Therefore, new strategies are needed to advance transvenous endovascular treatments by enhancing their safety and

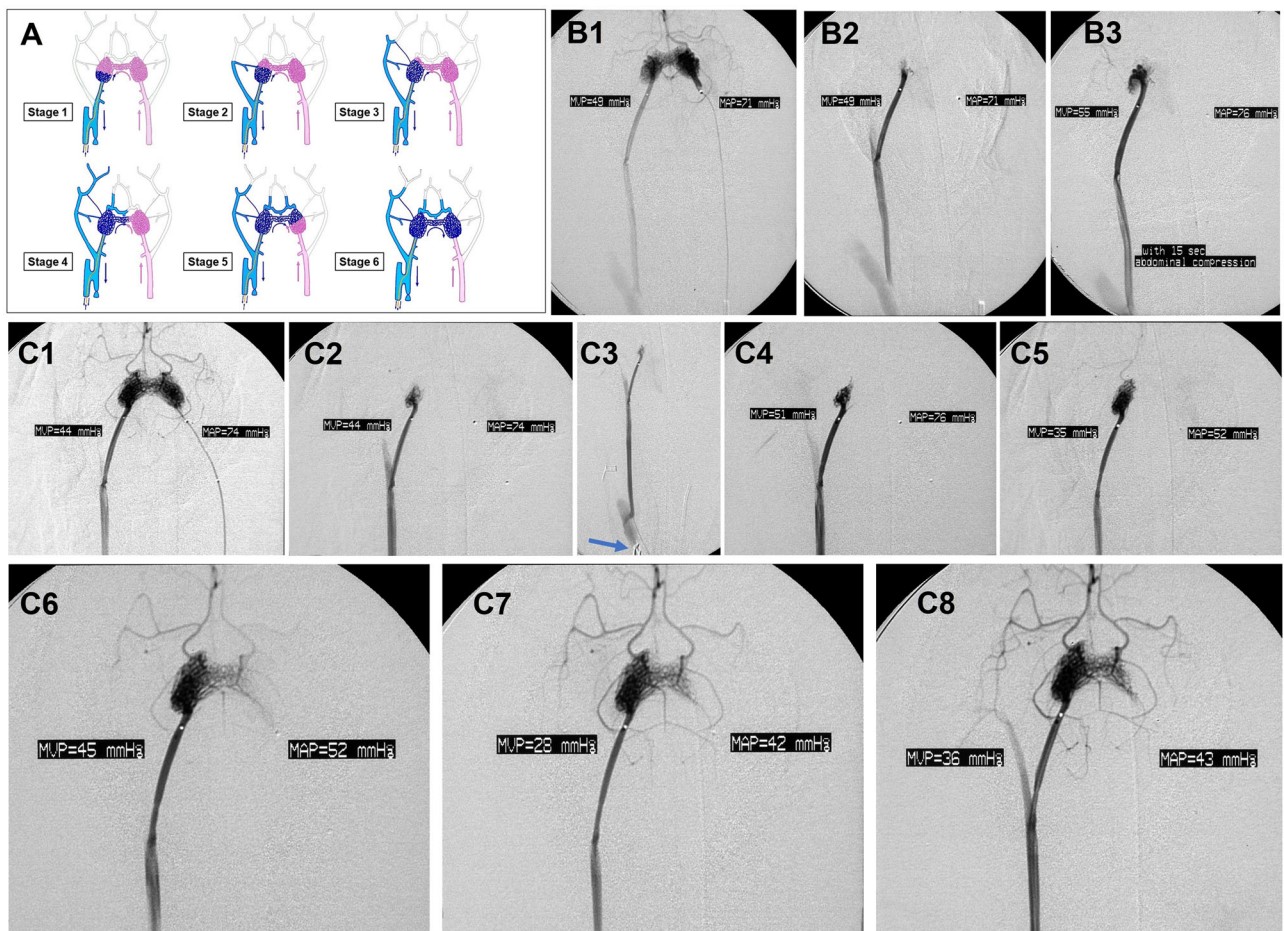

**Fig. 6 | Simulations of TRENSH with additional venous hypertension in the pig AVM model. A** Schematic diagrams of the pig AVM model and the seven "Stages" of retrograde nidus filling during simulations of TRENSH, as used by Massoud[16] Pink and blue arrows indicate direction of flow. Deeper levels of systemic and AF hypotension were related to progressively higher angiographic "Stages" signifying greater retrograde permeation of the nidus with contrast medium (blue) against the normal antegrade blood flow (pink). Blue curved arrow below nidus shows location of retrograde filling of nidus during TRENSH simulations. Some features of the original AVM model are depicted for consistency with previous accounts; in this study the right CC was patent and the right microcatheter did not straddle the fistula but was positioned in the right CC. **B** Attempted TRENSH simulation at systemic normotension plus venous hypertension (simulating raised CVP). Successive angiograms (**B1** antegrade injection via the AF, and **B2**, **B3** retrograde injections via the DV) showing minimal improvement in retropermeation of contrast medium, from a Stage 1 to a Stage 2 grade of nidus filling upon 15 s of abdominal compression. MAP: mean arterial pressure. MVP: mean venous pressure. **C** Attempted TRENSH simulation at systemic hypotension plus venous hypertension. Successive angiograms (**C1** antegrade injection via the AF, and **C2**, **C3** retrograde injections via the DV) showing venous clipping (blue arrow) alone caused minimal increased nidus retropermeation, but sequential clipping maneuvers worked in concert with progressive systemic hypotension to achieve near complete retrograde filling of the nidus. **C2** Transvenous angiogram at systemic normotension, with Stage 2 nidus filling; **C3** Same angiogram after clipping of right internal jugular vein shows mild improvement of retrograde nidus filling (Stage 3), detailed in (**C4**); **C5** Further improvement (Stage 4) after clip removed and systemic hypotension induced; **C6** When clip was reapplied there was further retropermeation (Stage 5); **C7** With progressive systemic hypotension, retropermeation remained at Stage 5; and **C8** With further clip application, there was a Stage 6 nidus filling.

efficacy; and several important factors must be considered. First, it is crucial to widen the scope and applications of transvenous treatment towards achieving cure in bAVMs >3 cm in size, that is, those higher Spetzler-Martin grade bAVMs that constitute the majority of bAVMs. At diagnosis, about 30% of bAVMs are <3 cm in size (small bAVMs), 60% are 3–6 cm (medium to large bAVMs), and 10% are >6 cm (large to giant bAVMs)[51]. Notably, larger bAVMs have lower mean AF pressures[52], which would be especially advantageous for two reasons: it would make it easier to reduce the transnidal pressure gradient to achieve greater nidus retropermeation, and the inherently lower transnidal gradient also would create less outflow into the DVs based on Darcy's law and Maag's formula[20]. Restrictively though, current transvenous treatments only target a small minority of (small) bAVMs, which are also selected because they tend to have one DV[2,10,49,53]. Second, we must aim to advance transvenous treatments for bAVMs with greater angioarchitectural complexity, which is usually the case in bAVMs >3 cm in size[9,11] These lesions usually have more than one DV; that is, more

than is acceptable for currently indicated transvenous embolization. Kellner et al. found that pediatric bAVMs had one DV when the mean Spetzler-Martin grade was 2.2, and had two or three DVs when the mean grade was 2.7[54]. Third, it is critical that we achieve less complication rates than is currently attainable, especially if treatment of larger, more complex bAVMs are to be attempted in the future. Despite knowing that multiple DVs (as in larger bAVMs) could in theory act as decompression valves to help avert raised intranidal pressures upon retrograde injection[16], this would apply only to use of liquid sclerosants as proposed in the original TRENSH concept. Unfortunately, the current use of viscous gel-like agents is still a problem in that regard because, with this embolic agent, a higher number of DVs increases the risk of inadvertent cast formation and occlusion of DVs (other than the one used for injection) before complete occlusion of the nidus, with consequent hemorrhagic complications, as pointed out by de Souza et al.[50] Use of viscous gel-like agents therefore dictates the requirement to choose only smaller bAVMs for treatment in order to control enough

nidus penetration and avoid the embolic agent prematurely and dangerously occluding other DVs that would be present in larger bAVMs. That balanced judgment may at times be difficult with a viscous embolic agent and could theoretically lead to increased rupture risk in nidus remnants following partial treatment[20].

In the present study, we conceive and demonstrate additional, previously untested theoretical TRENSH hemodynamic manipulations in the venous circulation downstream of bAVMs that provide greater knowledge and a foundation for further advances towards adoption of safe and effective TRENSH treatment for larger bAVMs, pending future optimization of an ideal safe and effective 'sclerosant' to use.

Limitations of this study include those of the experimental bAVM models, which have been discussed previously for biomathematical modeling[21,23], and for the pig AVM model[16,18,55]. Indeed, an important fundamental drawback inherent to all theoretical models is the absence of biological traits and natural biovariability—features that may also be incompletely captured in experimental in vivo systems. Several specific constraints of the present electrical model warrant acknowledgment, although these must be weighed against the many advantages it provides[23]. These limitations pertain to the imposed polarity of the nidus architecture, the absence of pulsatile blood flow, the lack of autoregulatory mechanisms, and the simplified representation of the AVM network. With respect to architectural polarity, the model assumes unidirectional flow from AFs on the left to DVs on the right, without recirculation or retrograde flow, whereas in vivo AFs and DVs may originate from or drain toward any region of the nidus. Regarding flow dynamics, the model employs Poiseuille's law, representing blood flow as steady through rigid conduits, despite some evidence that pulsatility—and the rate of increase in pulsatile pressure following therapeutic intervention—may contribute to risk of rupture. Autoregulation remains controversial—Nornes and Grip have argued that AVM vessels behave as fixed conduits incapable of autoregulatory responses[35], whereas Young et al. suggested the presence of some, albeit undefined, autoregulatory capacity[56]. Finally, although this network-based AVM model constitutes a more sophisticated representation than earlier approaches, it does not replicate the full structural complexity or heterogeneity of human AVMs. Nevertheless, it represents an important step toward more realistic patient specific computational modeling. Future work will require advanced simulations incorporating greater anatomical complexity and biological nuance, which may clarify the significance and influence of the factors we have discussed. This could also include modeling of the effects of prior multimodality treatments in conjunction with TRENSH[57]. Overall, while many of the theoretical results we obtained for comparisons between standard TRENSH versus venous-manipulation-augmented TRENSH were statistically significant, it remains to be seen in future clinical investigations if these proposed TRENSH innovations are of practical significance or if the attainable differences in real-world applications are too small to be meaningful. Therefore, future theoretical or clinically based simulations should also consider interpretation of modeling results that emphasize effect sizes as well as statistical significance. It should be noted that despite the apparent greater sophistication of the described theoretical bAVM model, a more simplified system might also provide meaningful future insights. For instance, this may be possible by using simplified mathematical/computational 1-D models of capillary circulations that can capture pressure drops along microvessels, flow distribution in branching microvascular networks, effects of viscosity changes, and resistance of capillary beds[58]. As such, these models may offer additional mathematically tractable, suitable methods for studying large networks of small vessels to help understand changes in the nidi of bAVMs. Finally, the small number of pigs we used for in vivo experiments is also a relative drawback but seems appropriate for our initial feasibility study of the outlined innovative strategies.

In conclusion, we demonstrate that mechanistically driven transnidal hemodynamic manipulations can enhance simulated transvenous endovascular treatments in experimental models of bAVMs. Specifically, strategies such as temporary CVP elevation, retrograde injection through dominant DVs, application of maximal safe transvenous injection pressures, and potentially, cardiac cycle-synchronized diastolic retroinjection of simulated sclerosant, can theoretically augment bAVM retropermeation during TRENSH simulations. These approaches show encouraging potential as adjunctive techniques for future clinical transvenous treatment of large and complex bAVMs. Our findings lay the groundwork for further validation studies to fully assess the translational potential of these theoretical strategies prior to clinical implementation.

## Data availability
The computational model datasets needed to reproduce the simulation figures are available in Figshare (https://doi.org/10.6084/m9.figshare.29143385). The raw outputs from each individual computational simulation are available from the corresponding author upon reasonable request. The data for the pig AVM model simulations are as presented in the "Results" section and the Supplementary Information.

## Code availability
The analysis code for algorithms of theoretical AVM model simulations is available at: https://github.com/kellenvu/massoud-avm.

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

## Acknowledgements

This research was partially funded by NIH grant 1R01HL052352-01A1 to support the pig experiments. Gratitude is extended to former colleagues and technical staff at the Rigler Radiological Research Center at UCLA for making the in vivo AVM model research possible, especially Cheng Ji and Fernando Viňuela. Thanks also to George Hademenos, formerly in the Department of Radiological Sciences at UCLA, and Mika Jain, formerly in the Departments of Physics and Computer Science at Stanford University for limited initial feasibility attempts at simulating systemic arterial hypotension manipulations for TRENSH in different theoretical AVM models than used in this research.

## Author contributions

T.F.M.: conceptualization, methodology, validation, formal analysis, writing—original draft preparation, writing—review and editing, visualization, supervision, approval of manuscript. B.V.: methodology, validation, formal analysis, writing—review and editing, visualization, approval of manuscript. K.V.: methodology, validation, formal analysis, writing—original draft preparation, writing—review and editing, visualization, approval of manuscript. J.J.H.: writing—review and editing, visualization, approval of manuscript. S.S.D.: conceptualization, methodology, writing—original draft preparation, writing—review and editing, visualization, approval of manuscript. All authors have read and agreed to the published version of the manuscript. Summary: T.F.M. and S.S.D. conceived different elements of the study. All authors contributed to the study design. Material preparation, data collection and analysis were performed by T.F.M., B.V., K.V., and S.S.D. The first draft of the manuscript was written by T.F.M., K.V., and S.S.D., and all authors commented on previous versions of the manuscript. All authors reviewed, edited, and approved the final manuscript.

## Competing interests

The authors have no relevant financial or non-financial interests to disclose. The authors have no competing interests to declare that are relevant to the content of this article. All authors certify that they have no affiliations with or involvement in any organization or entity with any financial interest or non-financial interest in the subject matter or materials discussed in this manuscript. The authors have no financial or proprietary interests in any material discussed in this article.
