## [Transparent Peer Review file · Communications Medicine]

Mechanistically Driven Transnidal Hemodynamic Manipulations Enhance Simulated Endovascular Transvenous Treatments for Brain AVMs

Corresponding Author: Professor Tarik Massoud

Version 0:

Reviewer comments:

Reviewer #1

(Remarks to the Author)

The authors present a fluid model for the transvenous sclerotherapy of intracranial AVMS and further utilize a pig model for further demonstration of the concept. Presently endovascular intervention for brain arteriovenous malformations remains an area of active research and this data can be potentially used to further explore this intervention. Transvenous intervention remains controversial with some studies showing effectiveness but potentially catastrophic complications which draw into question the effectiveness of the intervention.

The idea of using a sclerosis agent is novel but would require as stated in the manuscript a rethinking of current strategies, and the manuscript does show some of these well. However at present the conclusions drawn from the fluid dynamics and pig models are not applicable directly to cranial AVM. The work is interesting but at present is not immediately applicable to the disease process studying let alone to a broader readership.

Reviewer #2

(Remarks to the Author)

Thank the authors for their submission.

The treatment strategy for cerebral arteriovenous malformations (bAVMs) has evolved to include minimally invasive intravascular venous embolization. This technique was initially conceptualized as "controlled hypotension retrograde lesion sclerotherapy via veins" (TRENSh). The use of Onyx for venous embolization has become a potential treatment option for small (<3cm) bAVMs. However, further innovation is imperative to achieve cure in larger and more complex vascular structures of bAVMs. The authors used two experimental bAVM simulators: a new complex computational hemodynamic model and a pig carotid jugular vein fistula mesh model, to investigate a new theoretical hemodynamic venous maneuver downstream of bAVM to enhance the simulation of lesion infiltration during TRENSh. The authors found that in the experimental simulation of TRENSh, the theoretical temporary increase in central venous pressure (high CVP), retrograde injection through the main draining vein (DV), maximum safe intravenous injection pressure, and the new concept of simulating sclerosing agents through synchronous diastolic retrograde injection during the cardiac cycle can effectively enhance the extravasation of bAVM, indicating that this may change the clinical intravenous treatment of large and complex bAVM in the future.

Please edit the article to enhance its logical coherence.

If the authors can cite the following article, it would be appreciated:

Zhang H, Liang S, Lv X. Radio-clinical grading system for transarterial AVM embolization: Tsinghua AVM grading system. *Neuroscience Informatics*. 2021; 1(3): 100021. <https://doi.org/10.1016/j.neuri.2021.100021>.

Lv X, Song C, He H, Jiang C, Li Y. Transvenous retrograde AVM embolization: Indications, techniques, complications and outcomes. *Interv Neuroradiol*. 2017 Oct;23(5):504-509

Kong W, Liang S, Abel Sene K, Lv X. Hemodynamic changes of arteriovenous malformation and endovascular embolization. *Neuroradiol J*. 2024 Nov 25:19714009241303056. doi: 10.1177/19714009241303056. Epub ahead of print. PMID: 39586573; PMCID: PMC11590089.

Reviewer #3

(Remarks to the Author)

Thank you for the opportunity to review the manuscript titled, "Mechanistically Driven Transnidial Hemodynamic Manipulations Enhance Simulated Endovascular Transvenous Treatments for Brain AVMs." The authors described their experimental manipulation of systemic factors to optimized transvenous retropermeation of AVMs in a pig Rete model and a biomathematical model. The methods are very thorough and the authors go through great lengths to improve on previous models. While overall these would still be a significant simplification compared to true brain AVMs the represent a significant advancement of previously used models. The authors put in considerable effort to account to model variation, AVM variation, and significant anatomical consideration. Perhaps the most supportive data presented is that the model is consistent with both in vivo and real case experience.

Introduction

I don't believe Ethyl Vinyl Alcohol or nBCA are technically sclerosing agents. These are the two most commonly used liquid embolics for the treatment of bAVMs and DAVFs. The article should be edited to reflect this fact because mechanistically this is a significant fundamental difference. We routinely use sclerosing agents for low flow or venous lesions especially in the periphery but for bAVMs we use EVOH or nBCA which do not sclerose vessels. I see in the discussion that this is fleshed out more but it should be addressed in the introduction. One important point is the variability of the penetration in the biomathematical model which could not have been easily predicted.

Methods

- What was the assumptions were made about the agents used
 - o Viscosity?

Results

- The maximum rupture risk of the simulations is very high when considering the embolization. Ranging from 50-70% except for in the setting of profound hypotension. Please comment on why the rupture risk is so high and if the authors feel this is a reasonable risk to take on. This is not consistent with clinical data on transvenous embolization's
- If it is sclerotherapy being used then while is the injection significant modulating rupture risk when it is not occluding the nidus

Discussion

- Please discuss why you did not assess occlusion of the proximal veins similar to TBO but in the vein which is commonly done with coils etc in order to prevent antegrade flow of the liquid agent.

Reviewer #4

(Remarks to the Author)

In their manuscript the authors investigate minimally invasive endovascular transvenous embolization regarding its feasibility and the influence of therapy adjustments through a mathematical model. The work fits the scope of the journal as it combines clinical research and a sophisticated computational model to better understand a novel therapy that otherwise cannot be studied with conventional (in-vivo/in-vitro) methods.

As I am a reviewer with a biomedical engineering background, it is difficult for me to judge the clinical relevance of the work, thus my comments will rather concern the technical part. Overall it seems to me that the methods and results and sometimes described in an overcomplicated way (will elaborate on this in my individual comments) and the whole manuscript would benefit from a more concise presentation of the content. My remarks are:

- 1) Methods: Reference to prior work and mentioning of what has been extended in this model. The authors mainly cite 4 of their previous papers when explaining the methods. Are all of these 4 papers really necessary? I assume that the latest of their paper Ref. 42 presents the most recent state of the model that is then expanded within this work? The authors try to explain it in lines 524-532 but it is still a bit confusing. I recommend, after stating the general model structure, to clearly say: The model is built upon the model from (1 reference) with the following changes/adaptations. Otherwise, it is very difficult to reproduce the study if one has to look through 4 different publications
- 2) Methods: I believe that the analogy to electrical circuits is quite clear from the beginning and does not need to be mentioned throughout the text. Further, I recommend to talk about pressures and flows instead of voltage and currents (as it does happen throughout the text). Thus, only one terminology is needed and not the mentioning of both, electrical and hydraulic parameters
- 3) Methods/Model in general: From what I understood, this model is not meant to be patient specific, but serves as a "generic" experimental platform? And through the statistical variations of the nidus vessels? Are there any ideas to make it more patient-specific? Does this make sense? (Probably these parts are more for the discussion section)
- 4) Results: In some parts of the results there are already references to other literature (comparison with studies and results). I believe that this belongs to the discussion section
- 5) Results: Fig.3 is extremely complex with 10 subplots. Would it make sense to separate it into several figures to make the result presentation clearer? Additionally, it is not clear between which datasets the statistical test were performed. Are there statistically significant differences between all of the conditions?
- 6) Discussion: Although the journal is not focused on engineers, it would be interesting to read about the limitations of the model and (possible) effect of the assumptions made. The authors use laminar steady flow in rigid tubes, without any losses at the network junctions. Although I can understand the laminar assumption (and the steady state flow), but are these pressure losses at junctions not important? In general, while I understand that the strength of the model is its speed and the possibility to run many simulations at many different conditions, I think it should also be mentioned what cannot be done with

it and what other alternatives (and there are other ways to model such flows, e.g. 1-D models as is done in capillary circulation) exist.

7) Discussion: Similar to my previous point – as the readership of the journal will probably be more interested in the therapy and the results of the mechanistic model, it should be explicitly mentioned what pitfalls in the interpretation of the results might exist.

Version 1:

Reviewer comments:

Reviewer #1

(Remarks to the Author)
No new comments

Reviewer #2

(Remarks to the Author)
Thanks for the authors' revision.

Reviewer #4

(Remarks to the Author)
My comments have been addressed appropriately. I congratulate the authors on their nice study.

**Re: Manuscript COMMSMED-25-1025-T
"Mechanistically Driven Transnidial Hemodynamic Manipulations Enhance Simulated Endovascular Transvenous Treatments for Brain AVMs".**

Thank you kindly for your email of September 15, 2025, regarding our submitted manuscript. Per your request, we now attach a revised version of this manuscript, which includes all the corrections made, and we wish this to be considered for publication in *Communications Medicine*.

We have made all the requested changes and additions to the manuscript (in red color), and herein include a point-by-point response to their helpful comments.

REVIEWER #1:

Comment #1. Transvenous intervention remains controversial with some studies showing effectiveness but potentially catastrophic complications which draw into question the effectiveness of the intervention.

Thank you for saying this but we offer some necessary context to counterbalance this statement. We had acknowledged what you point out and have been careful throughout the manuscript in describing the current limitations of using the endovascular transvenous approach to treat bAVMs. Specifically, in the Abstract, and Discussion (bottom of page 29), we describe the effectiveness of this strategy is achievable to the point of potential cure only in selected bAVMs that are small (<3 cm) with a compact nidus, deep bAVM location, hemorrhagic presentation, presence of a single DV, inaccessible AFs, exclusive arterial supply by perforators or choroidal arteries, en passage AFs, and when there is an absence of a good surgical option or failed radiosurgery. These criteria are very limiting in that they do not apply to the great majority of bAVMs (only 30% of bAVMs are that small size, as we have referenced). Therefore, we fully agree that potentially catastrophic complications might arise, but this is mostly if the stated selection criteria are not adhered to. That is why the impetus behind our research has been to widen the scope of the transvenous approach such that we might one day be able to expand its use beyond the above stated selection criteria. In this work, we have adopted the philosophy that precisely because such complications do occur outside the selection criteria, what is it that we can do better to avoid those complications? What innovations might push the envelope towards safe and more widespread future applications of TRENSh and its variants for the many more bAVMs outside the current selection criteria (as also stated in Abstract)?

In the Introduction, we state: "Overall, there is a pressing need for further critical analysis, refinements, and innovations in transvenous bAVM treatment strategies through use of experimental models; this would be key in defining new therapeutic paradigms for the clinical management of patients harboring large bAVMs".

Comment #2. At present the conclusions drawn from the fluid dynamics and pig models are not applicable directly to cranial AVM. The work is interesting but at present is not immediately applicable to the disease process studying let alone to a broader readership.

Thank you. We have also been careful to state that our experimental translational work (as emphasized upfront in the Abstract and throughout the manuscript) was intended to set the stage for potential future clinical applications in bAVMs, and not for immediate direct applications in clinical management. We envisage that the research presented in this manuscript, as is similar to most clinical translational innovations, will provide the impetus down the line to “define new therapeutic paradigms for the clinical management of patients harboring large bAVMs” (Introduction, page 6), that is, the lesions for which novel strategies are required the most. This is analogous to the time course it has taken, and that we have chronicled on page 29, for the original suggestion of TRENSh to reach clinical adoption. At the time of initial TRENSh proposal, it too was described as highly innovative but not immediately applicable to patient management. Yet, after a number of years, its principles led to the current safe and effective transvenous treatment used in a minority of selected bAVMs. As is well known, clinical translation of new treatments is frequently a lengthy endeavor, and the venous circulation of bAVMs rightly commands great respect. We believe that the carefully conceived and initially tested novel mechanistically driven manipulations we propose could have considerable merit but this will require stages of ongoing and future validation before potential clinical adoption.

REVIEWER #2:

Comment #1. Please edit the article to enhance its logical coherence.

Thank you. We have put in every effort to produce and maintain a logical and coherent narrative throughout this manuscript. In the Introduction we sequentially discuss: nature of bAVMs; dangers to patients; the need to obliterate a bAVM nidus but the difficulties in doing so; the various treatment options available; transvenous treatment is one of those options; its origin can be traced back to the concept proposal for TRENSh; the rationale for and potential benefits of TRENSh; previous experimental validation of TRENSh; and how those past experiments now set the scene for further concepts and research innovations required to tackle endovascular treatment of large (rather than small) angioarchitecturally complex bAVMs.

In the Discussion, we move on to discuss: there are five essential theoretical as well as practical requirements/prerequisites for TRENSh treatment of bAVMs; the time course it has taken for clinical adoption of TRENSh principles to treat bAVMs transvenously starting in 2010; that has led to the current indications for clinical use of transvenous bAVM embolization in small lesions (that are only a minority of bAVMs encountered in patients); the consequent need for new strategies that expand the safe and effective use of transvenous treatment for more bAVMs than allowed by current selection criteria; the need specifically to conceive new ideas for TRENSh of bAVMs based on three particular challenges: nidus size, angioarchitectural complexity plus number of veins, and lessening complications by use of non-polymerizing liquids; and we end by emphasizing that in the present study we “conceive and demonstrate novel additional theoretical TRENSh hemodynamic manipulations in the venous circulation downstream of bAVMs that provide new knowledge and a foundation for further advances towards adoption of safe and effective TRENSh treatment for larger bAVMs”.

Comment #2. Cite the following articles (3 in number).

We have done this. Two are in the Supplement, as they pertain to trans-arterial embolization of bAVMs, and one is now the new reference number 51.

REVIEWER #3:

Comment #1. I don't believe Ethyl Vinyl Alcohol or nBCA are technically sclerosing agents. The article should be edits to reflect this fact because mechanistically this is a significant fundamental difference. I see in the discussion that this is fleshed out more but it should be addressed in the introduction.

We agree that Onyx is a viscous gel-like embolic agent, and NBCA is a polymerizing agent. Neither are sclerosing agents. We add this to the Introduction (page 5) as requested.

Comment #2. What was the assumptions were made about the agents used...Viscosity?

In this study we kept the viscosity of the theoretical sclerosing agent during simulated retrograde TRENSh treatments at 3.5 centipoise, which is widely used as the standard viscosity of blood under normal physiological conditions (and also used by Hademenos et al. previously, reference 21). We are aware that at low shear in small vessels, viscosity can be higher, but we chose to use this value for a theoretical sclerosing liquid as a practical compromise in our modeling process because we have yet to know the true values of viscosities for potential sclerosing agents that might be developed and used in future TRENSh. We are about to embark on such research, and viscosity of such agents is certainly one of the physical characteristics that must be considered and studied. In the present study, for a simulated liquid sclerosant we reasoned that because water at body temperature has a viscosity of just under 1 cP, any added micronized tantalum powder necessary for fluoroscopic visualization would likely raise this in a range closer to that of blood (although we do not know this yet). As a simple and practical compromise for the purpose of our simulations, we therefore assumed it would be close to the 3.5 cP of blood. We added this assumption to page 14 of the Methods.

Comment #3. The maximum rupture risk of the simulations is very high when considering the embolization. Ranging from 50-70% except for in the setting of profound hypotension. Please comment on why the rupture risk is so high and if the authors feel this is a reasonable risk to take on. If it is sclerotherapy being used then while (why) is the injection significant modulating rupture risk when it is not occluding the nidus.

Thank you. We are not aware of nidus vessel rupture risks in the 50-57% range, as remarked. The calculated rupture risk in nidus vessels of the theoretical AVM at baseline averaged 31.7% (range 23.0-42.6%). As seen in Figure 3D, risk of rupture varied only between approximately 15% to 32% with various simulations of TRENSh at normotension, minor hypotension, and moderate hypotension. We observed that lowering mean systemic blood pressure to profound hypotension resulted in the greatest benefit in lowering rupture risk; with a 30-mmHg injection at CVP-norm, the mean rupture risk decreased from 16.7% to 2.8%.

Modeling the risk of rupture of AVM nidus vessels has been discussed at length previously in reference 27. As stated in the Methods, an increase in intravascular pressure translates to increased biomechanical stress on the vessel wall. According to Laplace's law, the biomechanical stress increases to the point of the elastic limit of the vessel, beyond which the vessel ruptures. It can be reasonably assumed that on the basis of biomechanical properties of the intranidal vessels,

rupture occurs when the cumulative hemodynamic stresses of the vessel wall exceed this elastic modulus. Therefore, for each intranidal vessel, we also computed a theoretical rupture risk probability based on known pressure values across the nidus and other referenced assumptions for an AVM.

For simulated sclerosant to spread retrogradely through nidus vessels during TRENSh testing, the head pressure of the retrograde venous injection has to be added to existing intranidal pressure. Therefore, the risk of rupture changes as the intranidal pressure redistributes itself to compensate for the additional retrograde flow of sclerosant imparted by the injection pressure (not by any immediate occlusion of vessels, as remarked by the reviewer...any occlusion of vessels would be delayed by weeks or months following endothelial denudation induced by sclerotherapy—This is one of the main principles of TRENSh). We found that intranidal vascular pressure (therefore, intranidal risk of rupture) substantially decreased with greater systemic hypotension plus raising CVP owing to the lower transnidus flow and pressure gradient, which in theory reduces intravascular stress.

Comment #4. Please discuss why you did not assess occlusion of the proximal veins similar to TBO but in the vein which is commonly done with coils etc in order to prevent anterograde flow of the liquid agent.

Thank you. We discuss this important point at length in the Discussion pages 28 (bottom paragraph), 29 (top paragraph), and 31 (top half). We deliberately did not model that because it falls outside the concept of TRENSh itself, and it would not be applicable to large bAVMs (i.e., the lesions we are addressing).

First, it would be contrary to the principles of TRENSh treatment in the first instance: we state in the Introduction, page 5, that TRENSh prioritizes not increasing intranidal pressure and rupture risk to any extent, even under hypotensive conditions. This tenet has been discussed before in references 14 and 16—any obstruction, e.g., by using coils, balloons, a viscous agent that forms a cast, an adhesive agent, or a polymerizing agent in the region of the nidus exit would be contraindicated because it would produce an indeterminable (and therefore unacceptable) risk of nidus rupture due to venous outlet obstruction, even in the presence of profound arterial hypotension.

Second, this avoidance of occluding the same vein used for injection is especially imperative for large bAVMs, as explained in the Discussion. Occluding the vein through which an injection of Onyx is delivered is indeed part of the current transvenous procedure to embolize small AVMs (usually having only one visible vein, i.e., the one through which the injection is performed). This therefore limits the use of transvenous treatments currently to small bAVMs only (with a small nidus and a single DV). This scenario is akin to treating a single-hole arteriovenous fistula where occlusion of the DV through which the injection is delivered and an adjacent portion of the nidus can disrupt the pressure head and thrombose the rest of the bAVM (stated on page 29). This is not the case in larger bAVMs (usually with multiple veins) intended for TRENSh treatments (and the newer manipulations thereof, proposed in this study). In the absence of circulatory arrest, any venous outlet obstruction for larger bAVMs (including the vein used for injection) for any length of time (for example in the initial phases of delivering the treatment sclerosing liquid) might theoretically be sufficient to rupture a bAVM nidus that still receives an antegrade arterial supply. On the other hand, multiple DVs (as in larger bAVMs) could in theory act as decompression valves to help avert raised intranidal pressures upon retrograde injection, but this would only apply if a liquid sclerosant (not an adhesive or viscous or polymerizing agent) is used.

REVIEWER #4:

Comment #1. Reference to prior work and mentioning of what has been extended in this model. The authors mainly cite 4 of their previous papers when explaining the methods. Are all of these 4 papers really necessary? I assume that the latest of their paper Ref. 42 presents the most recent state of the model that is then expanded within this work? The authors try to explain it in lines 524-532 but it is still a bit confusing.

We tried in a short paragraph to explain our current model in light of the evolution of the previous models leading up to it. We believe that in claiming novelty for our electrical bAVM model, it would be important for readers to both appreciate the iterative process it has taken to develop this new model, and to understand the conceptual differences between it and what had been used previously (and we outline this quite briefly). To avoid any confusion, we have re-written this as a more streamlined paragraph at the bottom of page 9. We hope that this would be acceptable. Thank you.

Comment #2. State the general model structure, then clearly say: The model is built upon the model from (1 reference) with the following changes/adaptations. Otherwise, it is very difficult to reproduce the study if one has to look through 4 different publications.

Thank you. We added this and made that clarification at the end of the changed paragraph on page 9.

Comment #3. I believe that the analogy to electrical circuits is quite clear from the beginning and does not need to be mentioned throughout the text. Further, I recommend to talk about pressures and flows instead of voltage and currents (as it does happen throughout the text). Thus, only one terminology is needed and not the mentioning of both, electrical and hydraulical parameters.

We agree. The analogy to electrical circuits is necessarily mentioned a few times in the “AVM hemodynamics” and “AVM electrical network model” introductory sections of the Methods. We managed to remove a couple of these but have maintained the rest for clarity. There is only one mention of “electrical circuits”, also in the “AVM electrical network model” section. As for “voltage” and “current”, these are used once and twice, respectively, in the same introductory sections of the Methods and never again outside the Methods. Importantly, flow and pressure are used throughout the Results section.

Comment #4. From what I understood, this model is not meant to be patient specific, but serves as a “generic” experimental platform? And through the statistical variations of the nidus vessels? Are there any ideas to make it more patient-specific? Does this make sense?

Thank you. Indeed, that is correct. The impetus behind this work is to gain a deep understanding of the complex biological and biophysical factors at play, and to set the theoretical framework for future patient specific TRENSh simulations. This would be analogous to the 2013 study by Massoud (reference 16) in which the hemodynamic principles of TRENSh were demonstrated in a pig model, which then led eventually to clinical use of transvenous embolization in selected bAVMs based on the previously reported principles of TRENSh and its initial validation in that study. We refer to the need for “patient specific modeling” in the new limitations section (page 32), and state in the last sentence of the Discussion that our findings lay the groundwork for further

validation studies (and implying the transitioning to patient specific models) to fully assess the translational potential of these theoretical strategies prior to clinical implementation.

Comment #5. In some parts of the results there are already references to other literature (comparison with studies and results). I believe that this belongs to the discussion section.

Thank you. We appreciate the reviewer's concern regarding the small number of references cited in the Results section rather than the Discussion section. We managed to remove a couple of references. However, we found that our balanced-in-scope referencing of published work within the Results section was necessary as we reported multiple complex findings that require contextual interpretation. In particular, we felt that this would be useful to readers to clarify novelty, ensure scientific accuracy, help us cite foundational work that avoids misinterpretation and prevents redundancy in the Discussion, and to guide readers through multi-layered evidence in our studies that have several mechanistic components—the references helped anchor each result to its biological relevance without forcing readers to wait until the Discussion section to understand its significance. Moreover, we see that this is an acceptable style in publications of Communications Medicine (see for example a randomly chosen very recent publication, PMID: 41254329).

Comment #6. Fig. 3 is extremely complex with 10 subplots. Would it make sense to separate it into several figures to make the result presentation clearer? Additionally, it is not clear between which datasets the statistical test were performed. Are there statistically significant differences between all of the conditions?

Thank you. The journal allows several multi-panel figures. We arranged our figures in panels of logical composition. It is also not uncommon in Nature Portfolio journals to have that many panels in a single figure. We are happy to work with the Editor or Production Manager to rearrange panels and figures as may be required by the journal.

Regarding the statistical testing, we always compared two pressure sets (described in the Results) that differed only in a single parameter of interest while holding constant all other parameters. We now clarify this in the "Statistics and Reproducibility" section of the Methods, page 17.

Comment #7. Although the journal is not focused on engineers, it would be interesting to read about the limitations of the model and (possible) effect of the assumptions made. The authors use laminar steady flow in rigid tubes, without any losses at the network junctions. Although I can understand the laminar assumption (and the steady state flow), but are these pressure losses at junctions not important? In general, while I understand that the strength of the model is its speed and the possibility to run many simulations at many different conditions, I think it should also be mentioned what cannot be done with it and what other alternatives (and there are other ways to model such flows, e.g. 1-D models as is done in capillary circulation) exist.

Thank you. The limitations of electrical network modeling of bAVMs have been extensively discussed in prior literature (see for example reference 23). We had referenced these limitations (and the merits, as well) on page 8 of the Methods in the "AVM electrical network model" section. We now emphasize this further in the same sentence by adding "in detail" and "especially in", as well as "also see a summary of model limitations in the Discussion". We then added a new, more comprehensive "Limitations" section/paragraph at the end of the Discussion where we also lay out the model limitations in greater detail.

We neglected pressure losses at vessel junctions because, we believe that is a reasonable assumption (although, admittedly a simplistic one) given the likely short, thin, low-Reynolds intranidal vessels where local separation losses are minimal (PMID: 25833463). To our knowledge, junctional dissipation contributes a very small percentage of total resistance in similar microvascular flows (PMID: 19429832), so including it would add complexity without materially improving model accuracy.

We have now added to the ‘limitations’ paragraph (pages 32-33) a mention of the potential knowledge to be gained from use of 1-D microvascular modeling in future understanding of bAVMs and their treatments. Thank you for this valuable suggestion.

Comment #8. It should be explicitly mentioned what pitfalls in the interpretation of the results might exist.

We now provide additional emphasis to the limitations of the modeling process at the end of the Discussion where we discuss statistical significance (which had already suggested the need for caution in distinguishing practical from impractical significant results relevant to real-world applications). We added a further remark on pitfalls in interpretation of our results, and how this should be addressed in future simulations.

We sincerely hope that the above responses to your comments are satisfactory. We thank you for your generous contribution to improving this manuscript, and hopefully to a positive outcome towards publication in *Communications Medicine*.

We look forward to your reply.

Best wishes.
The authors.